# Accelerating RL for LLM Reasoning with Optimal Advantage Regression

**Kianté Brantley**[1]*, **Mingyu Chen**[2], **Zhaolin Gao**[3],
**Jason D. Lee**[4], **Wen Sun**[3], **Wenhao Zhan**[4], **Xuezhou Zhang**[2]
[1]Harvard University, [2]Boston University, [3]Cornell University, [4]Princeton University

## Abstract

Reinforcement learning (RL) has emerged as a powerful tool for fine-tuning large language models (LLMs) to improve complex reasoning abilities. However, state-of-the-art policy optimization methods often suffer from high computational overhead and memory consumption, primarily due to the need for multiple generations per prompt and the reliance on critic networks or advantage estimates of the current policy. In this paper, we propose $A^\star$-PO, a novel two-stage policy optimization framework that directly approximates the optimal advantage function and enables efficient training of LLMs for reasoning tasks. In the first stage, we leverage offline sampling from a reference policy to estimate the optimal value function $V^\star$, eliminating the need for costly online value estimation. In the second stage, we perform on-policy updates using a simple least-squares regression loss with only a single generation per prompt. Theoretically, we establish performance guarantees and prove that the KL-regularized RL objective can be optimized without requiring complex exploration strategies. Empirically, $A^\star$-PO achieves competitive performance across a wide range of mathematical reasoning benchmarks, while reducing training time by up to $2\times$ and peak memory usage by over 30% compared to PPO, GRPO, and `REBEL`[Gao et al., 2024a]. Implementation of $A^\star$-PO can be found at https://github.com/ZhaolinGao/A-PO.

## 1 Introduction

Recent advances in large language models (LLMs), including OpenAI-o1 [OpenAI, 2024], DeepSeek-R1 [DeepSeek-AI, 2025], and Kimi-1.5 [Team, 2025], have demonstrated remarkable reasoning capabilities. These models excel at producing long Chain-of-Thought (CoT)[Wei et al., 2023, DeepSeek-AI, 2025, Zeng et al., 2025] responses when tackling complex tasks and exhibit advanced, reflection-like reasoning behaviors[Gandhi et al., 2025]. A key factor driving these improvements is reinforcement learning (RL) with rule-based rewards derived from ground-truth answers [Lambert et al., 2025, DeepSeek-AI, 2025, Team, 2025], where models receive binary feedback indicating whether their final answers are correct.

Substantial efforts have been devoted to refining RL algorithms, such as PPO [Schulman et al., 2017] and GRPO [Shao et al., 2024], to further improve performance and training stability [Richemond et al., 2024, Wang et al., 2024, Ji et al., 2024, Liu et al., 2024, Yu et al., 2025, Liu et al., 2025b]. However, these methods either require explicit critic networks to estimate value functions or advantages, or rely on multiple generations per prompt, leading to substantial computational overhead and memory consumption. These limitations make it challenging to scale to long-context reasoning tasks and larger model sizes. This naturally raises the question: ***Can we develop simpler and more efficient RL algorithms for long context reasoning?***

---

*Authors are listed in alphabetical order of their last names.
Correspondence to `mingyuc@bu.edu`, `zg292@cornell.edu`, `wenhao.zhan@princeton.edu`

39th Conference on Neural Information Processing Systems (NeurIPS 2025).

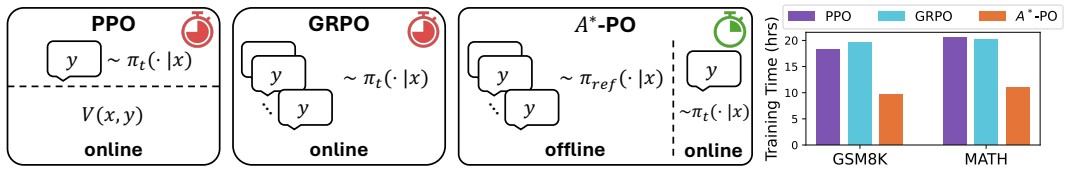

Figure 1: We present $A^\star$-PO, an efficient, regression-based approach for LLM post-training. Prior methods such as GRPO and PPO incur high computational costs, either due to requiring multiple samples per prompt or maintaining an explicit value network. In contrast, $A^\star$-PO simplifies the training process by estimating the optimal value function using offline generations from $\pi_{\mathsf{ref}}$ and requiring only a single response per prompt during online RL. As a result, $A^\star$-PO reduces training time by up to $2\times$ compared to GRPO and PPO.

Our answer is $A^\star$-PO, Policy Optimization via Optimal Advantage Regression, a policy optimization algorithm that uses ***only a single sample*** per prompt during online RL. Instead of relying on an explicit value network or multiple online generations to estimate the advantage of the current policy during training, our approach directly approximates the fixed **optimal** value function in an offline manner. We observe that the value function of the optimal policy of the KL-regularized RL can be expressed as an expectation under the reference policy. Based on this insight, the first stage of the algorithm performs *offline* sampling from the reference policy to estimate the optimal values for all prompts in the training set. This stage is highly parallelizable and can efficiently leverage fast inference libraries without requiring gradient computations. In the second stage, we perform on-policy updates via regressing optimal advantages with least square regression, where the advantages are constructed using the optimal values from the first stage. The on-policy updates only use a single generation per prompt, which drastically reduces both the computational and memory overhead associated with RL training. Thus, algorithmically, $A^\star$**-PO eliminates heuristics such as clipping and response-wise reward normalization, resulting an extremely simple algorithm**.

Theoretically, we establish formal performance guarantees for $A^\star$-PO, showing that it achieves near-optimal performance. Notably, our theoretical analysis reveals that, **without sophisticated exploration, $A^\star$-PO learns a near-optimal policy with polynomial sample complexity, as long as the base model's probability of solving a math question is lower bounded above zero**. Experimentally, we evaluate $A^\star$-PO extensively across a range of reasoning benchmarks, including GSM8K [Cobbe et al., 2021], MATH [Hendrycks et al., 2021], and competition-level datasets such as AMC, AIME, and HMMT. Across multiple model sizes, including Qwen2.5-1.5B, 3B, and 7B, our approach consistently achieves comparable or superior results to strong baselines such as PPO, GRPO, and REBEL [Gao et al., 2024a], while achieving the lowest KL-divergence to the base model, and reducing training time by up to $2\times$ and peak memory usage by over 30%.

## 2 Preliminary

In this work, we denote $x$ as a prompt (e.g., a math question) and $y$ as a generation (e.g., a reasoning chain plus a solution for a given math question). Consider the following KL-regularized RL objective:

$$\max_\pi \mathbb{E}_{x, y \sim \pi(\cdot|x)} r(x, y) - \beta \mathsf{KL}(\pi(\cdot|x)|\pi_{\mathsf{ref}}(\cdot|x)).$$

We mainly consider the setting where $r$ is binary: $r(x, y) = 1$ if the generation $y$ contains the correct answer and $0$ otherwise. It is well known that the optimal policy is $\pi^\star(y|x) \propto \pi_{\mathsf{ref}}(y|x) \exp(r(x, y)/\beta)$. We also denote $V^\star(x)$ as the optimal value function of the KL-regularized RL objective above. It can be shown that $V^\star$ has the following closed-form expression:

$$\forall x: \ V^\star(x) = \beta \ln \mathbb{E}_{y \sim \pi_{\mathsf{ref}}(\cdot|x)} \left[ \exp(r(x, y)/\beta) \right]. \tag{1}$$

Note that the expectation in $V^\star$ is under $\pi_{\mathsf{ref}}$, which indicates a simple way of estimating $V^\star$: we can generate multiple i.i.d. responses from $\pi_{\mathsf{ref}}$ and take the average to estimate $V^\star$. We will show that under realistic conditions, the bias and variance of such an estimator are well under control. We aim to learn a policy that can maximize the expected reward (e.g., maximize the average accuracy). Particularly, our theoretical results will model the performance gap between our learned policy and $\pi^\star$, the optimal policy of the KL-regularized RL objective, in terms of maximizing the expected reward.

---
**Algorithm 1** $A^\star$-PO: Policy Optimization via Optimal Advantage Regression
---
**Require:** Training prompts set $\mathcal{X}$, reference policy $\pi_{\mathsf{ref}}$, temperature $\beta$, sample size $N$, iterations $T$
**Ensure:** Learned policy $\pi_T$

1: ### Stage 1: Estimating $V^\star(x)$ for all training prompts $x$ in $\mathcal{X}$
2: **for all** $x \in \mathcal{X}$ **do**
3:     Draw $N$ i.i.d. samples $\{y_i\}_{i=1}^N \sim \pi_{\mathsf{ref}}(\cdot \mid x)$
4:     Compute $\widehat{V}^\star(x) \leftarrow \beta \ln\left(\frac{1}{N} \sum_{i=1}^N \exp\left(r(x, y_i)/\beta\right)\right)$.
5: **end for**

6: (Optional: eliminate all training prompts whose N solutions are all wrong)
7: ### Stage 2: On-policy Update with one rollout per prompt
8: **for** $t = 1$ **to** $T$ **do**
9:     Collect a batch of training samples $\mathcal{D} := \{(x, y)\}$ with $x \in \mathcal{X}$ and $y \sim \pi_t(\cdot|x)$
10:    Update policy $\pi_t$ to $\pi_{t+1}$ by performing SGD on the following least square loss $\ell_t(\pi)$

$$\ell_t(\pi) := \sum_{(x,y) \in \mathcal{D}} \left(\beta \ln \frac{\pi(y|x)}{\pi_{\mathsf{ref}}(y|x)} - \left(r(x,y) - \widehat{V}^\star(x)\right)\right)^2$$

11: **end for**
---

## 3 Algorithm

The algorithm consists of two stages. The first stage collects data from $\pi_{\mathsf{ref}}$ and estimates $V^\star(x)$ for all $x$ in the training set. Particularly, for every $x$ in the training set, we generate $N$ i.i.d responses $y_1, \ldots, y_N$, and estimate $V^\star(x)$ as: $\widehat{V}^\star(x) = \beta \ln \left(\sum_{i=1}^N \exp(r(x, y_i)/\beta)/N\right)$. Note that in general $\widehat{V}^\star$ is a biased estimate of $V^\star$ due to the nonlinearity of the $\ln$ function. However, we will show that as long as $v_{\mathsf{ref}}(x)$, the probability of $\pi_{\mathsf{ref}}$ generating a correct solution at $x$, is lower bounded above zero, then the bias $|\widehat{V}^\star(x) - V^\star(x)|$ can be well controlled and shrinks quickly as $N$ increases. We emphasize that while this stage requires generating $N$ responses per prompt $x$ from $\pi_{\mathsf{ref}}$, **it can be done completely offline, the data can be collected efficiently in parallel without any gradient computation, and any off-shelf faster inference library can be used for this stage.**

The second stage performs the on-policy update. Specifically, at iteration $t$, given the current policy $\pi_t$, we optimize the following loss:

$$\ell_t(\pi) := \mathbb{E}_{x, y \sim \pi_t(\cdot|x)} \left(\beta \ln \frac{\pi(y|x)}{\pi_{\mathsf{ref}}(y|x)} - \left(r(x,y) - \widehat{V}^\star(x)\right)\right)^2,$$

where the expectation is taken under the current policy $\pi_t$ and $r(x, y) - \widehat{V}^\star(x)$ approximates the optimal advantage $A^\star(x, y) = r(x, y) - V^\star(x)$ in the bandit setting as $Q(x, y) = r(x, y)$—hence the name $A^\star$-PO. To estimate the least-squares loss $\ell_t(\pi)$, we collect a batch of online samples $\{(x, y)\}$ with $y \sim \pi_t(\cdot|x)$ and perform a small number of stochastic gradient descent steps, starting from $\pi_t$, to obtain the updated policy $\pi_{t+1}$ as outlined in Algorithm 1. **Notably, during this online stage, we generate only a single sample $y$ per prompt $x$, significantly accelerating the training.**

The motivation behind the above loss is that when $\widehat{V}^\star(x) = V^\star(x)$, the optimal policy $\pi^\star$ for the KL-regularized RL objective is the global minimizer of the least-squares regression loss, regardless of the distribution under which the expectation is defined (i.e., MSE is always zero under $\pi^\star$). Unlike popular RL algorithms such as PPO [Schulman et al., 2017] and GRPO [Shao et al., 2024], we do not apply a clipping mechanism to constrain $\pi_{t+1}$ from deviating too far from $\pi_t$. Instead, we introduce a single regularization term based on the KL divergence to the fixed reference policy $\pi_{\mathsf{ref}}$. Furthermore, while PPO and GRPO rely on the *advantage of the current policy*, defined as $A^{\pi_t} := r(x, y) - V^{\pi_t}(x)$, we instead use the *optimal advantage* $A^\star$. A key benefit of using $V^\star$ is that it can be pre-computed efficiently using a large number of samples from $\pi_{\mathsf{ref}}$ (Eq. (1)), whereas $V^{\pi_t}$ must be estimated on the fly, requiring either multiple generations during training (e.g., GRPO or RLOO [Kool et al., 2019]) or maintaining an explicit critic network (e.g., PPO). Since $A^\star$-PO only generates one response per prompt, it also eliminates the heuristic of response-wise reward

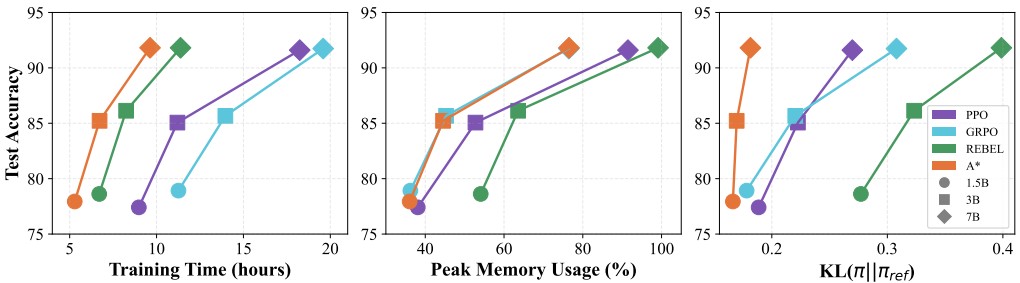

Figure 2: **Test accuracy versus training time, peak memory usage, and KL divergence** across four baselines and three model sizes on GSM8K. Our approach (orange) can achieve comparable performance (accuracy) to baselines GRPO and PPO, while being 2x faster, more memory efficient, and achieving a smaller KL divergence. Note that for $A^\star$-PO, the training time includes the time from **both** stages (i.e., offline data collection from $\pi_{\mathsf{ref}}$ and online RL training).

normalization. Finally, most existing policy optimization algorithms—including PG [Williams, 1992], TRPO [Schulman et al., 2015], GRPO, PPO, and REBEL [Gao et al., 2024a]—follow the idea of approximate *policy iteration* [Bertsekas and Tsitsiklis, 1996, Kakade and Langford, 2002] where $\pi_{t+1}$ is designed to approximately optimize the advantage function of the current policy $A^{\pi_t}$, subject to an implicit or explicit constraint that prevents it from deviating too far from $\pi_t$. The consequence of relying on such an approximate policy iteration style update is that it requires additional strong structural condition such as policy completeness [Bhandari and Russo, 2024]. In contrast, $A^\star$-PO does not follow the approximate policy iteration paradigm. It places no explicit constraints on keeping $\pi_{t+1}$ close to $\pi_t$. Instead, it directly aims to learn $\pi^\star$ by regressing on the optimal advantages.

## 4 Experiments

Our implementation of $A^\star$-PO closely follows the pseudocode in Algorithm 1, with the only modification of using two different KL-regularization coefficients, $\beta_1$ and $\beta_2$, during stages 1 and 2 respectively. In stage 1, we employ a relatively large $\beta_1$ to ensure a smoother estimation of $V^\star(x)$. In contrast, a smaller $\beta_2$ is used in stage 2 to relax the KL constraint to $\pi_{\mathsf{ref}}$, encouraging the learned policy $\pi$ to better optimize the reward. Although this introduces an additional hyperparameter, we find that the same set of $\beta_1$ and $\beta_2$ works well across different datasets and model sizes, and therefore, we keep them fixed throughout all experiments. In stage 1, we sample $N = 8$ responses per prompt to estimate the optimal value function with $\beta_1 = 1/2$. At stage 2, we collect a dataset $\mathcal{D} = (x, y)$ with $x \in \mathcal{X}$ and $y \sim \pi_t(\cdot \mid x)$, and optimize the least-squares regression objective using gradient descent with AdamW [Loshchilov and Hutter, 2017] and $\beta_2 = 1e - 3$. We empirically evaluate the performance of $A^\star$-PO across various reasoning datasets, model sizes, and evaluation benchmarks. Additional details are provided in Appendix A, and qualitative analysis is shown in Appendix B.

### 4.1 Basic Reasoning on GSM8K

**Training Details.** We conduct experiments on the GSM8K dataset [Cobbe et al., 2021], which consists of grade school-level math problems. We compare $A^\star$-PO against several baseline RL algorithms, including PPO [Schulman et al., 2017], GRPO [Shao et al., 2024], and REBEL [Gao et al., 2024a]. Simpler policy gradient methods such as REINFORCE with Leave-One-Out (RLOO) [Kool et al., 2019] are not included, as prior work has shown that REBEL consistently outperforms RLOO in performance with similar computational efficiency [Gao et al., 2024a]. We sample two generations per prompt for GRPO and use a model of the same size as the policy for the critic in PPO. Following the DeepSeek-R1 training recipe [DeepSeek-AI, 2025], we perform RL directly on the base model without any prior supervised fine-tuning (SFT). We report results using three different model sizes—1.5B, 3B, and 7B—based on the pre-trained Qwen2.5 models [Qwen, 2025] with a maximum context length of $1,024$. A rule-based reward function is used, assigning +1 for correct answers and 0 for incorrect ones. All experiments are implemented using the VERL framework [Sheng et al., 2025].

| Model Size | Method | Time (hrs) | Memory (%) | KL ($\pi\|\|\pi_{\text{ref}}$) | MATH500 | Minerva Math | Olympiad Bench | AMC 23 Avg@32 |
|---|---|---|---|---|---|---|---|---|
| 1.5B | *base* | / | / | / | 45.8 | 16.91 | 17.66 | 20.55 |
| | PPO | 10.84 | 42.64 | 0.151 | 57.0 | 21.69 | 20.92 | 29.84 |
| | GRPO | 15.01 | 42.19 | 0.091 | **58.0** | **22.79** | 21.07 | **32.19** |
| | REBEL | 10.33 | 63.34 | 0.098 | 57.8 | 20.22 | **23.15** | 31.67 |
| | $A^\star$-PO | **6.92** | **41.59** | **0.069** | 57.8 | 22.43 | 22.26 | 31.88 |
| 3B | *base* | / | / | / | 50.8 | 23.16 | 23.74 | 29.77 |
| | PPO | 13.59 | 54.95 | 0.111 | 65.8 | 23.90 | 26.71 | 34.68 |
| | GRPO | 18.26 | 49.75 | 0.102 | 66.0 | 25.00 | **28.93** | 34.61 |
| | REBEL | 12.88 | 74.32 | 0.099 | **67.0** | **27.57** | 28.19 | **36.33** |
| | $A^\star$-PO | **8.78** | 49.28 | **0.082** | 66.2 | 25.74 | 28.04 | 35.47 |
| 7B | *base* | / | / | / | 62.8 | 22.43 | 30.56 | 39.61 |
| | PPO | 20.53 | 92.81 | 0.133 | 74.4 | 30.88 | 33.98 | 55.47 |
| | GRPO | 20.15 | 77.82 | 0.172 | 73.2 | 33.46 | 33.98 | 55.86 |
| | REBEL | 14.67 | 98.77 | 0.124 | 74.6 | **34.56** | **34.72** | **56.88** |
| | $A^\star$-PO | **11.01** | **76.57** | **0.078** | **76.2** | **34.56** | 34.27 | 56.25 |

Table 1: **Results on MATH.** For each metric, the best-performing method is highlighted in **bold**, and the second-best is underlined.

**Evaluation.** We evaluate each method based on its trade-off between accuracy and KL-divergence on the validation set, assessing the effectiveness of each algorithm in optimizing the KL-regularized RL objective. Additionally, we report the peak GPU memory usage (as a percentage of total available memory) during backpropagation, as well as the total training time (in hours) for each method. Both runtime and memory usage are measured using 4 H100 GPUs under the same hyperparameter settings detailed in Appendix A.5. *The training time for $A^\star$-PO also includes the offline generation time in stage 1*. Peak memory usage is averaged over 100 batches.

**$A^\star$-PO is faster, more memory-efficient, and better optimizes the KL-constrained RL objective.** Figure 2 presents the training time, peak memory usage, and KL divergence to $\pi_{\text{ref}}$ across four methods and three model sizes. While all methods achieve similar test accuracy, they exhibit substantial differences in the other three metrics. Although REBEL achieves comparable training time as $A^\star$-PO, it requires significantly more memory due to processing two generations simultaneously during each update. GRPO shows similar peak memory usage but requires two generations per prompt and performs twice as many updates as $A^\star$-PO, leading to approximately $2\times$ longer training time. PPO incurs both higher computational cost and memory usage due to its reliance on an explicit critic. By consistently updating with respect to $\pi_{\text{ref}}$, $A^\star$-PO maintains the smallest KL divergence to the reference policy. ***Overall, $A^\star$-PO achieves the fastest training time, lowest peak memory usage, and smallest KL divergence compared to all baselines with similar test accuracy.***

## 4.2 Advanced Reasoning on MATH

**Training Details.** In this section, we evaluate on the more advanced MATH dataset [Hendrycks et al., 2021], which consists of challenging problems from high school math competitions. Following the original experimental setup, we use 7,500 questions for training and randomly select 1,000 questions for validation. We adopt the same model and hyperparameter settings as the previous section.

**Evaluation.** Following prior work [Zeng et al., 2025], we evaluate model performance on standard mathematical reasoning benchmarks, including MATH500 [Hendrycks et al., 2021], Minerva Math [Lewkowycz et al., 2022], and OlympiadBench [He et al., 2024], as well as the competition-level AMC 2023 benchmark. For AMC 2023, due to the small size of the benchmark (40 questions), we report average accuracy over 32 generations to reduce variance.

**$A^\star$-PO achieves higher accuracy, is the fastest and generalizes effectively across benchmarks.** Table 1 presents the results, where the best-performing method is highlighted in **bold** and the second-best is underlined. As shown, $A^\star$-PO consistently ranks as either the best or second-best method across various benchmarks and model sizes, while being faster than baselines, especially PPO and GRPO, when model size is large (e.g., 2x faster than PPO and GRPO for 7B model size). $A^\star$-PO is also the most memory efficient and achieves the smallest KL divergence to the base mdoel $\pi_{\text{ref}}$. For in-distribution test set MATH500, $A^\star$-PO achieves similar performance to baselines on smaller

| Method | AIME 24 | | AIME 25 | | HMMT Feb 24 | | HMMT Feb 25 | | Average | |
|---|---|---|---|---|---|---|---|---|---|---|
| | Avg@32 | Pass@32 | Avg@32 | Pass@32 | Avg@32 | Pass@32 | Avg@32 | Pass@32 | Avg@32 | Pass@32 |
| *base* | 21.25 | 60.00 | 21.15 | 43.33 | 9.48 | 36.67 | 8.85 | 30.00 | 15.21 | 42.50 |
| PPO | **30.94** | **80.00** | 25.12 | **46.67** | 15.94 | **43.33** | **14.06** | **50.00** | 21.52 | **55.00** |
| GRPO | 30.00 | 73.33 | 25.00 | 43.33 | 15.63 | 40.00 | 13.23 | 40.00 | 20.97 | 49.17 |
| $A^\star$-PO | 29.17 | 70.00 | **26.67** | **46.67** | **16.77** | 43.33 | 13.65 | 43.33 | **21.57** | 50.83 |

Table 2: **Long-context reasoning results using the DeepSeek Distilled 1.5B Model.** For each metric, the best-performing method is highlighted in **bold**.

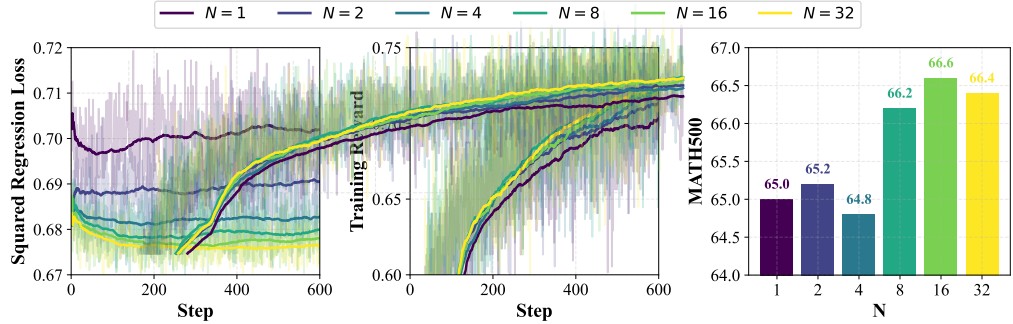

Figure 3: **Ablation results with different number of $N$ for estimating $V^\star$.** Solid lines indicate the moving average with window size 100. (Left) Squared regression loss per step of $A^\star$-PO. (Middle) Training reward per step. (Right) Model performance on MATH500 with varying values of $N$.

model sizes (1.5B and 3B), but outperforms on the larger model size (7B). One may wonder if explicitly pre-computing $V^\star$ for all training prompts would result overfitting to the training set. When evaluating on the out-of-domain benchmarks, $A^\star$-PO also demonstrates strong generalization capabilities.

### 4.3   Long-context Reasoning with DeepSeek-distilled Qwen

**Training Details.** In this section, we assess the ability of $A^\star$-PO to train with long context lengths. We use the DeepSeek-R1 distilled Qwen-1.5B model [DeepSeek-AI, 2025] with a context length of 16,384, training on a randomly selected subset of 5,000 problems from the VGS-AI/OpenR1-Cleaned dataset [Wang et al., 2025]. We limit training to a subset of the dataset due to the large scale of the original corpus and the significant computational cost of data generation and training on academic hardware. We exclude results for REBEL, as it runs out of memory under this long-context setting. The same hyperparameter configurations from the previous section are used.

**Evaluation.** We evaluate on competition-level AIME and HMMT benchmarks from 2024 and 2025, reporting both Avg@32 and Pass@32 metrics.

**$A^\star$-PO outperforms baselines and effectively scales to long-context reasoning.** The results are presented in Table 2, with the best-performing method highlighted in bold. $A^\star$-PO achieves the highest performance on AIME 2025 and HMMT February 2024, and performs comparably to GRPO and PPO on AIME 2024 and HMMT February 2025. Averaged over these 4 math competition benchmarks, $A^\star$-PO achieves the best performance (measured under the metric of Avg@32). These results demonstrate that estimating $V^\star$ using 8 generations remains effective even in long-context reasoning scenarios, and that $A^\star$-PO can be successfully applied to such settings. Unsurprisingly, $A^\star$-PO remains the most efficient method, requiring only 49 hours of training time, compared to 88 hours for PPO and 90 hours for GRPO.

### 4.4   Ablations

**Number of Generations.** We conduct an ablation on the number of generations $N$ used during stage 1 to estimate $V^\star$ on the MATH dataset with the 3B model, varying $N$ from 1 to 32. The results are shown in Figure 3. We observe that the squared regression loss consistently decreases as $N$ increases,

indicating that larger values of $N$ lead to more accurate estimations of $V^\star$. The accuracy of this estimation significantly affects both the training reward and downstream evaluation metrics, such as performance on MATH500. Notably, the training reward increases almost monotonically with larger $N$, and a similar trend is observed for performance on MATH500, which begins to plateau at $N = 8$. We provide a discussion of the plateau behavior in Appendix C.

**Filtering Out Hard Problems.** For fair comparison to the baselines, our main result did not filter out hard questions where pass@N is zero. In this section, we explore whether training efficiency can be further improved by filtering out problems whose pass@N is zero. Note that pass@N can be easily estimated using the N samples generated from phase 1 of $A^\star$-PO. The intuition is that problems the reference policy $\pi_{\mathsf{ref}}$ is unable to solve (i.e., with near-zero success probability) are unlikely to be solved through RL post-training. Our method naturally supports this filtering mechanism, as we already estimate $V^\star$ during the offline stage. We conduct an ablation study varying $N$ from 2 to 32, filtering out problems where all $N$ sampled responses are incorrect. The results are shown in Figure 4. Filtering significantly reduces training time by reducing the effective size of the training set, as illustrated by the green bars compared to the dashed baseline. When $N$ is small, filtering removes more problems since it is more likely that none of the $N$ responses are correct, further accelerating training. In terms of performance,

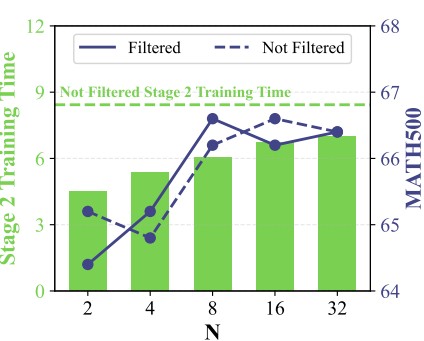

Figure 4: **Ablation Results on Filtering Hard Prompts.** (Green) Training time of Stage 2 for different values of $N$. The dashed line indicates the training time without filtering. (Purple) Performance on MATH500 with and without filtering.

the MATH500 evaluations for filtered and unfiltered runs remain closely aligned. Notably, the filtered run with $N = 8$ achieves the highest accuracy, even surpassing the unfiltered baseline while reducing training time by 28%.

$\beta_1$ **for Estimating $V^\star$.** The parameter $\beta_1$ controls the level of smoothness when estimating $V^\star$. As $\beta_1 \to 0$, the estimated $V^\star$'s value approaches Pass@N (computed using the collected $N$ responses), while as $\beta_1 \to \infty$, it estimates $V^{\pi_{\mathsf{ref}}}$. Proofs of these limiting cases are provided in Appendix D. We ablate $\beta_1$ from $\infty$ to $1/8$, and the results are shown in Figure 5. From the squared regression loss, we observe that smaller values of $\beta_1$ lead to higher initial training loss. But the final training loss decreases monotonically as $\beta_1$ decreases, until reaching $\beta_1 = 1/8$. A similar trend is observed for both the training reward and MATH500 performance, which improve monotonically with smaller $\beta_1$ values up to $\beta_1 = 1/8$. Further decreasing $\beta_1$ may produce an overly optimistic estimate of $V^\star$, which the model cannot realistically achieve, ultimately hindering performance.

## 5  Theory

In this section, we present a theoretical analysis of a more general variant of Algorithm 1 and demonstrate that the output policy achieves near-optimal expected performance against $\pi^\star$, the optimal policy of the KL-regularized RL objective, with polynomial complexity. Surprisingly, we will see that under reasonable conditions of $\pi_{\mathsf{ref}}$, we can achieve the above goal *without explicit exploration*. Denote $p(x)$ as the probability of $\pi_{\mathsf{ref}}$ returning a correct solution and assume $\min_x p(x) \geq v_{\mathsf{ref}} > 0$, the key message of this section is that:

> **KL-regularized RL with $\beta > 0$ and $v_{\mathsf{ref}} > 0$ can be solved without any sophisticated exploration (e.g., optimism in the face of uncertainty).**

We demonstrate the above key message via a reduction to no-regret online learning. Specifically, we assume access to a *possibly stochastic* no-regret oracle NR. In each iteration, the oracle is invoked with the historical estimated losses $\widehat{\ell}_{1:t-1}$ and policy class $\Pi$ to compute a distribution $q_t \in \Delta(\Pi)$ over policies for the next iteration: $q_t = \mathsf{NR}(\widehat{\ell}_{1:t-1}, \Pi)$. We then sample a policy $\pi_t \sim q_t$ and use it to generate a prompt-response pair where $x_t \sim \rho$ and $y_t \sim \pi_t(\cdot|x_t)$.[2] Here $\rho$ is the distribution

---

[2]For simplicity, we just assume at each iteration $t$, our on-policy batch only contains one $(x_t, y_t)$ pair.

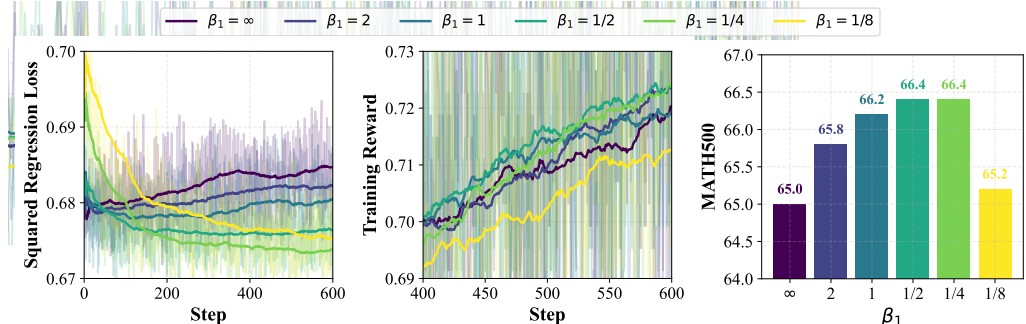

Figure 5: **Ablation results with different $\beta_1$ for estimating $V^\star$.** Solid lines indicate the moving average with window size 100. (Left) Squared regression loss per step of $A^\star$-PO. (Middle) Training reward per step. (Right) Model performance on MATH500 with varying values of $\beta_1$.

of the prompts in the dataset. Given the pair $(x_t, y_t)$, we construct the following **least square loss** $\widehat{\ell}_t$:

$$\widehat{\ell}_t(\pi) := \left( \beta \ln \frac{\pi(y_t|x_t)}{\pi_{\text{ref}}(y_t|x_t)} - \left( r(x_t, y_t) - \widehat{V}^\star(x_t) \right) \right)^2.$$

Algorithm 1 indeed is a specific instance of the above algorithm that leverages online gradient descent (**OGD**) [Hazan et al., 2016] as the oracle NR. We use $\text{Reg}(T)$ to denote the cumulative regret of NR over $T$ iterations:

$$\text{Reg}(T) := \sum_{t=1}^{T} \mathbb{E}_{\pi \sim q_t} \left[ \widehat{\ell}_t(\pi) \right] - \min_{\pi \in \Pi} \sum_{t=1}^{T} \widehat{\ell}_t(\pi).$$

**Remark.** When the policy class and loss functions are convex, deterministic no-regret oracles such as OGD and mirror descent [Hazan et al., 2016] can guarantee a regret bound of $\text{Reg}(T) = O(\sqrt{T})$. In more general settings, stochastic oracles like follow-the-perturbed-leader (**FTPL**) [Suggala and Netrapalli, 2020] are also capable of achieving $\sqrt{T}$ regret.

Due to the possible randomness of NR, we will utilize the decoupling coefficient (DC), a standard complexity measure in contextual bandits [Zhang, 2022, 2023, Li et al., 2024], to characterize the difficulty of learning over the policy class $\Pi$:

**Definition 1.** *The decoupling coefficient* $\text{DC}(\Pi)$ *is defined as the smallest* $d > 0$ *such that for all distribution* $q$ *on* $\Pi$*:*

$$\mathbb{E}_{\pi \sim q, x, y \sim \pi(\cdot|x)}[\ln \pi(y|x) - \ln \pi^\star(y|x)] \leq \sqrt{d \cdot \mathbb{E}_{\pi \sim q, \pi' \sim q, x, y \sim \pi'(\cdot|x)}[(\ln \pi(y|x) - \ln \pi^\star(y|x))^2]}.$$

DC is perhaps the most general structural condition known in the contextual bandit and RL theory literature, and is known to be much more general than other popular measures such as Eluder dimension [Zhang, 2023] which cannot easily capture non-linear structures in the problems.

Our analysis relies on some standard assumptions. We first assume that the policy class is realizable.

**Assumption 1.** *Suppose that* $\pi^\star \in \Pi$*.*

Next, we assume the loss function is bounded.

**Assumption 2.** *Suppose that* $\|r_\pi\|_\infty \leq C_l$ *where* $r_\pi(x, y) := \beta \ln \frac{\pi(y|x)}{\pi_{\text{ref}}(y|x)}$ *for all* $\pi \in \Pi$*.*

The above assumption mainly ensures that our predictor $\ln \pi/\pi_{\text{ref}}$ has bounded outputs, a standard assumption that is used in least squares regression analysis. These assumptions are commonly adopted in the policy optimization literature as well [Rosset et al., 2024, Xie et al., 2024, Gao et al., 2024a,b, Huang et al., 2024].

Recall that our loss uses $\widehat{V}^\star$ to approximate $V^\star$. We introduce a simple and reasonable condition which can ensure the bias $|\widehat{V}^\star - V^\star|$ is small.

**Assumption 3.** *Suppose that we have $\mathbb{E}_{y \sim \pi_{\text{ref}}(\cdot|x)}[r(x, y)] \geq v_{\text{ref}} > 0$ for all $x$.*

Assumption 3 is practically reasonable, which intuitively says that we hope $\pi_{\text{ref}}$ has non-zero Pass@k value at $x$. For example, consider $v_{\text{ref}} = 0.05$. One can show that with probability greater than 90%, $\pi_{\text{ref}}$'s Pass@50 on $x$ is one. In practice, we typically do not expect RL post-training can solve problems whose Pass@50 under $\pi_{\text{ref}}$ is zero anyway.

Now we are ready to present a bound on the performance gap between the output policies and $\pi^\star$.

**Theorem 1.** *Suppose that Assumptions 1 to 3 hold true. With probability at least $1 - \delta$, we have*

$$\mathbb{E}_{x,y \sim \pi^\star} r(x, y) - \frac{1}{T} \sum_{t=1}^{T} \mathbb{E}_{\pi \sim q_t}[\mathbb{E}_{x,y \sim \pi} r(x, y)] \lesssim \left( \frac{\mathsf{DC}(\Pi) \mathsf{Reg}(T)}{T\beta^2} \right)^{\frac{1}{4}} + \left( \frac{\mathsf{DC}(\Pi) C_l^2 \ln(1/\delta)}{T\beta^2} \right)^{\frac{1}{4}}$$

$$+ \left( \left( \min \left\{ \exp\left(\frac{1}{\beta}\right) - 1, \frac{1}{v_{\text{ref}}} \right\} \right)^2 \frac{\mathsf{DC}(\Pi) \ln(|\mathcal{X}|/\delta)}{N} \right)^{\frac{1}{4}}. \quad (2)$$

Theorem 1 shows that the performance gap decays with rates $O\left(\mathsf{Reg}(T)^{\frac{1}{4}} T^{-\frac{1}{4}}\right)$ and $O\left(N^{-\frac{1}{4}}\right)$, implying that a near-optimal policy can be learned with polynomial sample complexity as long as $\mathsf{Reg}(T)$ is sublinear. Based on this theoretical guarantee, our algorithm exhibits two notable advantages over existing provable online policy optimization methods:

- **Only realizability.** Our approach optimizes sequence of loss functions constructed from on-policy samples. Unlike other on-policy policy optimization approaches (e.g., PG [Bhandari and Russo, 2024] and its variants PPO [Schulman et al., 2017], REBEL [Gao et al., 2024a]) which often relies on a stronger condition called policy completeness, our approach relies on a much weaker realizability condition.
- **No exploration.** Our method does not incorporate any explicit exploration mechanism such as optimism in the face of uncertainty or reset to some exploratory distribution. As a result, it is conceptually simpler and more computationally efficient. For instance, the loss function $\widehat{\ell}_t$ is just a least square regression loss. Treating $\ln \pi$ as a predictor, then the loss $\widehat{\ell}_t$ is a convex functional with respect to $\ln \pi$ as a whole. In contrast, prior work that relies on explicit exploration (e.g., optimism in the face of uncertainty) requires bi-level optimization which often leads to non-convex loss functionals [Cen et al., 2024, Xie et al., 2024].

**Case study: log-linear policies.** We present the learning guarantee for log-linear policies as a special case where we show that our simple regression loss indeed can be optimized via simple SGD. Consider the following log-linear policy class:

$$\Pi = \left\{ \pi_\theta : \pi_\theta(y|x) = \frac{\exp\left(\langle \theta, \phi(x, y) \rangle\right)}{\sum_{y'} \exp\left(\langle \theta, \phi(x, y') \rangle\right)}, \quad \theta \in \Theta \right\}, \quad (3)$$

where $\Theta \subseteq \mathbb{R}^d$ is the parameter space and $\phi : \mathcal{X} \times \mathcal{Y} \to \mathbb{R}^d$ represents predefined feature vectors. First, instantiating the learning oracle NR as **FTPL** [Suggala and Netrapalli, 2020], we can achieve sublinear regret when learning log-linear policies:

**Corollary 1** (log-linear policies with FTPL, *Informal*). *Consider the log-linear policy class $\Pi$ defined in Eq. (3). Suppose Assumptions 1 to 3 hold true and use FTPL as the NR oracle. Then we have the following regret with probability at least $1 - \delta$:*

$$\mathbb{E}_{x,y \sim \pi^\star} r(x, y) - \frac{1}{T} \sum_{t=1}^{T} \mathbb{E}_{\pi \sim q_t}[\mathbb{E}_{x,y \sim \pi} r(x, y)] \lesssim \left( \frac{d^{\frac{5}{2}} C_l}{T^{\frac{1}{2}} \beta} \right)^{\frac{1}{4}} + \left( \frac{d C_l^2 \ln(1/\delta)}{T\beta^2} \right)^{\frac{1}{4}}$$

$$+ \left( \left( \min \left\{ \exp\left(\frac{1}{\beta}\right) - 1, \frac{1}{v_{\text{ref}}} \right\} \right)^2 \frac{d \ln(|\mathcal{X}|/\delta)}{N} \right)^{\frac{1}{4}}.$$

Alternatively, we can select **OGD** as NR, following the practical implementation in Algorithm 1. In this case, our analysis requires an additional assumption, which states that the Fisher information matrix remains full-rank throughout the algorithm's runtime.

**Assumption 4.** *Suppose there exists $\lambda > 0$ such that we have $A(\theta_t) \succeq \lambda I$ for all $t$ where $A(\theta)$ is the Fisher information matrix:*

$$A(\theta) = \mathbb{E}_{x,y\sim\pi_\theta(\cdot|x)}[\phi(x,y)\phi(x,y)^\top] - \mathbb{E}_{x,y\sim\pi_\theta(\cdot|x)}[\phi(x,y)]\mathbb{E}_{x,y\sim\pi_\theta(\cdot|x)}[(\phi(x,y))^\top].$$

Under Assumption 4, we can show that OGD is able to learn $\pi^\star$ with a $O(T^{-\frac{1}{2}})$ convergence rate:

**Theorem 2** (log-linear policies with OGD, *Informal*). *Consider the log-linear policy class $\Pi$ defined in Eq. (3). Suppose Assumptions 1, 2 and 4 hold true and use OGD as the NR oracle. Then we have with probability at least $1 - \delta$ that:*

$$\mathbb{E}_{x,y\sim\pi^\star} r(x,y) - \mathbb{E}_{x,y\sim\pi_{T+1}}[r(x,y)] \lesssim \left( \frac{B^2}{T} + \frac{C_l^2 \ln(T/\delta)}{\beta^2\lambda^2 T} + \frac{C_l B \ln(1/\delta)}{\beta\lambda T} \right)^{\frac{1}{2}}.$$

This theorem suggests that Algorithm 1 has three key strengths over existing policy optimization methods [Gao et al., 2024a, Cen et al., 2024, Xie et al., 2024] for log-linear policies:

- **Last-iterate convergence.** Unlike existing methods that output a mixture of all the online policies, Algorithm 1 ensures that the final policy $\pi_{T+1}$ is close to $\pi^\star$, which is more favorable in practice.
- **Dimension-free learning rate.** The learning rate of Algorithm 1 is independent of the parameter space dimension $d$, making it scalable for high-dimensional function classes.
- **Computational efficiency.** OGD is computationally efficient and trackable in practice.

More details are presented in Appendices E to G.

## 6 Related Work

**RL for LLM Reasoning.** Reinforcement learning (RL) has become a standard for post-training LLMs, including Reinforcement Learning from Human Feedback (RLHF) [Christiano et al., 2017, Touvron et al., 2023] and Reinforcement Learning with Verifiable Rewards (RLVR) [DeepSeek-AI, 2025]. Notable examples include GRPO [Shao et al., 2024], DR-GRPO [Liu et al., 2025a], DAPO [Liu et al., 2024], OREO [Wang et al., 2024], DQO [Ji et al., 2024], and RAFT++ [Xiong et al., 2025]. Similar to OREO, DAPO, DQO, and DRO, $A^\star$-PO operates efficiently by requiring only a single response per prompt and regressing the log ratio of $\pi$ and $\pi_{\text{ref}}$ to the advantage. However, unlike these methods, $A^\star$-PO eliminates the need to train an explicit critic for value estimation, further improving its computational efficiency.

**Exploration in RL.** To combat the slow convergence and suboptimal policies yielded by classic entropy-based methods [Schulman et al., 2017, Shao et al., 2024], recent work has studied active exploration [Xiong et al., 2024, Ye et al., 2024, Xie et al., 2024], which offers stronger theoretical guarantees but often suffers from practical instability. In contrast, motivated by recent findings [Zhao et al., 2025, Yue et al., 2025] that exploring beyond the base model's knowledge is frequently wasteful—and that existing RL methods struggle to outperform the base model's pass@K performance—, $A^\star$-PO is designed for a more realistic setting, where $\pi_{\text{ref}}$ has non-zero pass@K (for some reasonably large K) on the target prompts. We show that in this setting, active exploration is unnecessary: $A^\star$-PO provably converges to $\pi^\star$ with polynomial sample complexity, achieving both efficiency and meaningful optimality. Additional discussions of the related works are provided in Appendix H.

## 7 Conclusion

We introduced $A^\star$-PO, a simple and efficient policy optimization algorithm that directly regresses the estimated optimal advantage function, removing the need for critics and multiple generations per prompt. It also completely removes heuristics such as clipping and trajectory reward normalization. $A^\star$-PO enables faster and more memory-efficient training, achieves smaller KL-divergences to the base model, and consistently achieves competitive or superior performance on reasoning benchmarks, with strong theoretical guarantees.

## Acknowledgements

KB acknowledge: This work has been made possible in part by a gift from the Chan Zuckerberg Initiative Foundation to establish the Kempner Institute for the Study of Natural and Artificial Intelligence. ZG is supported by LinkedIn under the LinkedIn-Cornell Grant. WS acknowledges fundings from NSF IIS-2154711, NSF CAREER 2339395, DARPA LANCER: LeArning Network CybERagents, Infosys Cornell Collaboration, and Sloan Research Fellowship.

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

# A Experiment Details

## A.1 Dataset Details

Table 3: Dataset split, maximum prompt length, and maximum generation length

| Dataset | Huggingface Dataset Card | Train - Val | Prompt Length | Generation Length |
|---|---|---|---|---|
| GSM8K | openai/gsm8k | 7.47k - 1.32k | 256 | 1,024 |
| MATH | xDAN2099/lighteval-MATH | 7.5k - 1k | 256 | 1,024 |
| OpenR1-Math-220K | open-r1/OpenR1-Math-220k | 5k - / | 256 | 16,384 |

Table 4: Dataset prompt format

| Dataset | Prompt Format |
|---|---|
| GSM8K | **{prompt}** Let's think step by step and output the final answer after "####". |
| MATH | **{prompt}** Let's think step by step and output the final answer within \boxed{}. |
| OpenR1-Math-220K | <\|begin_of_sentence\|><\|User\|>**{prompt}**<\|Assistant\|><think> |

## A.2 Model Details

For GSM8K and MATH, we use Qwen2.5 model series with size 1.5B (model card: Qwen/Qwen2.5-1.5B), 3B (model card: Qwen/Qwen2.5-3B), and 7B (model card: Qwen/Qwen2.5-7B) with **full parameter** training on 4 H100 GPUs. For OpenR1, we use DeepSeek-distilled Qwen model (model card: deepseek-ai/DeepSeek-R1-Distill-Qwen-1.5B) with **full parameter** training on 8 H100 GPUs. For PPO, we use the same model as the policy for the critic.

## A.3 Reward Details

We use a rule-based reward function based solely on the correctness of the response, assigning +1 for correct answers and 0 for incorrect ones. Recent studies [Chen et al., 2025b] have proposed incorporating format-based rules into reward calculations to encourage models to follow specific output formats. However, in our experiments, we observed no significant difference in performance with or without such format-based rewards. Therefore, for simplicity, we exclude them from our implementation.

## A.4 KL and Entropy penalty

For PPO, to ensure that the learned policy $\pi$ remains close to the reference policy $\pi_{\text{ref}}$, an additional KL penalty is applied to the reward:

$$r(x, y) - \gamma_{\text{KL}} \left( \ln \pi(y \mid x) - \ln \pi_{\text{ref}}(y \mid x) \right), \tag{4}$$

where $r(x, y)$ is the rule-based reward for prompt $x$ and response $y$, and $\gamma_{\text{KL}}$ controls the strength of the KL penalty. For GRPO, we adopt the same KL penalty as in [Shao et al., 2024], also parameterized by $\gamma_{\text{KL}}$. To further encourage exploration in PPO and GRPO, we apply standard entropy regularization by subtracting the policy entropy, computed over the generated responses in the batch, from the loss. This term is weighted by a coefficient $\gamma_{\text{entropy}}$. *Note that A\*-PO does not require these additional penalties, highlighting the simplicity of our algorithm.*

## A.5   Hyperparameter Details

| Setting | Parameters | |
|---|---|---|
| Generation (train) | temperature: 1.0 | top p: 1 |
| Generation (validation) | temperature: 0 | |
| PPO | batch size: 256 | $\gamma_{\text{entropy}}$: 1e-3 |
| | mini batch size: 128 | $\gamma_{\text{KL}}$: 1e-4 |
| | micro batch size: 1 | gae $\gamma$: 1 |
| | policy learning rate: 1e-6 | gae $\lambda$: 1 |
| | critic learning rate: 1e-5 | clip ratio: 0.2 |
| | train epochs: 25 | |
| GRPO | batch size: 256 | $\gamma_{\text{entropy}}$: 1e-3 |
| | mini batch size: 128 | $\gamma_{\text{KL}}$: 1e-4 |
| | micro batch size: 1 | num rollout: 2 |
| | policy learning rate: 1e-6 | |
| | critic learning rate: 1e-5 | |
| | train epochs: 25 | |
| REBEL | batch size: 256 | $\gamma_{\text{entropy}}$: 0 |
| | mini batch size: 128 | $\gamma_{\text{KL}}$: 0 |
| | micro batch size: 2 | num rollout: 2 |
| | policy learning rate: 1e-6 | $\eta$: 1e3 |
| | critic learning rate: 1e-5 | |
| | train epochs: 25 | |
| $A^{\star}$-PO | batch size: 256 | $\beta_1$: 1/2 |
| | mini batch size: 128 | $\beta_2$: 1e-3 |
| | micro batch size: 1 | |
| | policy learning rate: 1e-6 | |
| | critic learning rate: 1e-5 | |
| | train epochs: 25 | |

## A.6   Evaluation Details

We evaluate each method based on the trade-off between accuracy and KL divergence on the test set, assessing the effectiveness of each algorithm in optimizing the KL-regularized RL objective. Additionally, we report the peak GPU memory usage (as a percentage of total available memory) during backpropagation, as well as the total training time (in hours) for each method. Both runtime and memory usage are measured using 4 H100 GPUs under the same hyperparameter settings detailed in Appendix A.5. *The reported training time for $A^{\star}$-PO includes the offline generation time in stage 1.* Peak memory usage is averaged over 100 batches.

Following prior work [Zeng et al., 2025], we evaluate model performance on standard mathematical reasoning benchmarks, including MATH500 [Hendrycks et al., 2021], Minerva Math [Lewkowycz et al., 2022], and OlympiadBench [He et al., 2024], as well as competition-level benchmarks such as AMC 2023 and AIME/HMMT from 2024 and 2025. For competition-level benchmarks, due to their small size (30–40 questions), we report average accuracy over 32 generations to reduce evaluation variance.

For benchmarks other than competition-level benchmarks, we use greedy decoding with temperature set to 0. For competition-level benchmarks, we sample with a temperature of 0.6 and a top-$p$ value of 0.95.

# B    Qualitative Analysis

In this section, we qualitatively demonstrate that, after training with $A^\star$-PO, the model exhibits cognitive behaviors that are essential for effective reasoning, including verification, subgoal setting and backtracking, as identified by Gandhi et al. [2025].

## B.1    Verification

**Prompt:** If $-6 \le a \le -2$ and $3 \le b \le 5$, what is the greatest possible value of $\left(a + \frac{1}{b}\right)\left(\frac{1}{b} - a\right)$? Express your answer as a common fraction. Let's think step by step and output the final answer within \boxed{}.

---

**Qwen2.5-7B Generation:** To find the greatest possible value of the expression $\left(a + \frac{1}{b}\right)\left(\frac{1}{b} - a\right)$, we can start by simplifying the expression. Notice that the expression is a difference of squares:

$$\left(a + \frac{1}{b}\right)\left(\frac{1}{b} - a\right) = \left(\frac{1}{b}\right)^2 - a^2 = \frac{1}{b^2} - a^2$$

Given the ranges for $a$ and $b$: - $-6 \le a \le -2$ - $3 \le b \le 5$
To maximize $\frac{1}{b^2} - a^2$, we need to maximize $\frac{1}{b^2}$ and minimize $a^2$.
1. The maximum value of $\frac{1}{b^2}$ occurs when $b$ is at its minimum value, which is 3. So, $\frac{1}{b^2} = \frac{1}{3^2} = \frac{1}{9}$.
2. The minimum value of $a^2$ occurs when $a$ is at its maximum value, which is -2. So, $a^2 = (-2)^2 = 4$.
Therefore, the greatest possible value of the expression is:

$$\frac{1}{9} - 4 = \frac{1}{9} - \frac{36}{9} = \frac{1 - 36}{9} = \frac{-35}{9}$$

So, the greatest possible value of the expression is $\boxed{-\dfrac{35}{9}}$.

---

**Qwen2.5-7B after $A^\star$-PO Generation:** To find the greatest possible value of the expression $\left(a + \frac{1}{b}\right)\left(\frac{1}{b} - a\right)$, we start by simplifying it. Notice that the expression is in the form of a difference of squares:

$$\left(a + \frac{1}{b}\right)\left(\frac{1}{b} - a\right) = \left(\frac{1}{b}\right)^2 - a^2 = \frac{1}{b^2} - a^2.$$

Given the ranges for $a$ and $b$:

$$-6 \le a \le -2 \quad \text{and} \quad 3 \le b \le 5,$$

we need to maximize $\frac{1}{b^2} - a^2$.
1. The term $\frac{1}{b^2}$ is maximized when $b$ is minimized. Since $3 \le b \le 5$, the minimum value of $b$ is 3. Therefore, the maximum value of $\frac{1}{b^2}$ is $\frac{1}{3^2} = \frac{1}{9}$.
2. The term $-a^2$ is maximized when $a^2$ is minimized. Since $-6 \le a \le -2$, the minimum value of $a^2$ is $(-2)^2 = 4$. Therefore, the maximum value of $-a^2$ is $-4$.
Combining these, the greatest possible value of the expression is:

$$\frac{1}{9} - 4 = \frac{1}{9} - \frac{36}{9} = -\frac{35}{9}.$$

However, we need to check if this is indeed the maximum value by considering the boundary conditions. Let's verify by calculating the expression at the boundary values:
1. When $a = -6$ and $b = 3$:

$$\left(-6 + \frac{1}{3}\right)\left(\frac{1}{3} + 6\right) = \left(-\frac{17}{3}\right)\left(\frac{19}{3}\right) = -\frac{323}{9}.$$

2. When $a = -2$ and $b = 3$:

$$\left(-2 + \frac{1}{3}\right)\left(\frac{1}{3} + 2\right) = \left(-\frac{5}{3}\right)\left(\frac{7}{3}\right) = -\frac{35}{9}.$$

3. When $a = -6$ and $b = 5$:

$$\left(-6 + \frac{1}{5}\right)\left(\frac{1}{5} + 6\right) = \left(-\frac{29}{5}\right)\left(\frac{31}{5}\right) = -\frac{899}{25}.$$

4. When $a = -2$ and $b = 5$:

$$\left(-2 + \frac{1}{5}\right)\left(\frac{1}{5} + 2\right) = \left(-\frac{9}{5}\right)\left(\frac{11}{5}\right) = -\frac{99}{25}.$$

From these calculations, we see that the greatest possible value is indeed achieved when $a = -2$ and $b = 3$, giving us:

$$\boxed{-\dfrac{35}{9}}.$$

In this example, Qwen2.5-7B after $A^\star$-PO exhibits verification behavior where it performs systematic error-checking after arriving at the answer.

## B.2 Subgoal Setting

**Prompt:** Let $a$ and $b$ be positive real numbers such that $ab^2 = 5$. Find the minimum value of $a^4 + b^8$. Let's think step by step and output the final answer within \boxed{}.

---

**Qwen2.5-7B Generation:** To find the minimum value of $a^4 + b^8$ given that $ab^2 = 5$, we can use the method of Lagrange multipliers or substitution. Here, we will use substitution.
Given:
$$ab^2 = 5$$
We want to minimize:
$$f(a, b) = a^4 + b^8$$
First, express $a$ in terms of $b$ using the constraint:
$$a = \frac{5}{b^2}$$
Substitute this into the function $f(a, b)$:
$$f(b) = \left(\frac{5}{b^2}\right)^4 + b^8$$
$$f(b) = \frac{625}{b^8} + b^8$$
Now, we need to find the critical points by taking the derivative of $f(b)$ with respect to $b$ and setting it to zero:
$$f'(b) = -\frac{5000}{b^9} + 8b^7$$
Set the derivative equal to zero:
$$-\frac{5000}{b^9} + 8b^7 = 0$$
$$8b^{16} = 5000$$
$$b^{16} = \frac{5000}{8}$$
$$b^{16} = 625$$
$$b = \sqrt[16]{625}$$
$$b = 5^{1/4}$$
Now, substitute $b = 5^{1/4}$ back into the expression for $a$:
$$a = \frac{5}{(5^{1/4})^2}$$
$$a = \frac{5}{5^{1/2}}$$
$$a = 5^{1/2}$$
Now, substitute $a = 5^{1/2}$ and $b = 5^{1/4}$ into the function $f(a, b)$:
$$f(5^{1/2}, 5^{1/4}) = (5^{1/2})^4 + (5^{1/4})^8$$
$$f(5^{1/2}, 5^{1/4}) = 5^2 + 5^2$$
$$f(5^{1/2}, 5^{1/4}) = 25 + 25$$
$$f(5^{1/2}, 5^{1/4}) = 50$$
Thus, the minimum value of $a^4 + b^8$ is $\boxed{50}$.

**Qwen2.5-7B after $A^\star$-PO Generation:** To find the minimum value of $a^4 + b^8$ subject to the constraint $ab^2 = 5$, we can use the method of substitution. From the constraint, we can express $a$ in terms of $b$:

$$a = \frac{5}{b^2}.$$

Now, substitute this expression for $a$ into the function $a^4 + b^8$:

$$a^4 = \left(\frac{5}{b^2}\right)^4 = \frac{625}{b^8}.$$

So, the expression $a^4 + b^8$ becomes:

$$\frac{625}{b^8} + b^8.$$

Let's denote $x = b^8$. Then, the expression becomes:

$$\frac{625}{x} + x.$$

We need to find the minimum value of this expression. To do this, we can take the derivative with respect to $x$ and set it to zero:

$$\frac{d}{dx}\left(\frac{625}{x} + x\right) = -\frac{625}{x^2} + 1.$$

Setting the derivative equal to zero, we get:

$$-\frac{625}{x^2} + 1 = 0 \implies x^2 = 625 \implies x = 25 \quad (\text{since } x = b^8 > 0).$$

Now, we need to verify that this critical point is a minimum by checking the second derivative:

$$\frac{d^2}{dx^2}\left(\frac{625}{x} + x\right) = \frac{1250}{x^3} + 1.$$

At $x = 25$, the second derivative is:

$$\frac{1250}{25^3} + 1 = \frac{1250}{15625} + 1 = \frac{100}{1250} + 1 = \frac{2}{25} + 1 = \frac{2}{25} + \frac{25}{25} = \frac{27}{25} > 0,$$

which confirms that $x = 25$ is a minimum. Therefore, the minimum value of $\frac{625}{x} + x$ occurs at $x = 25$. Substituting $x = 25$ back into the expression, we get:

$$\frac{625}{25} + 25 = 25 + 25 = 50.$$

Finally, we need to express $b$ in terms of $x$. Since $x = b^8$, we have $b^8 = 25$, so $b = 25^{1/8}$. And since $a = \frac{5}{b^2}$, we have $a = \frac{5}{(25^{1/8})^2} = \frac{5}{25^{1/4}} = \frac{5}{5^{1/2}} = \frac{5}{\sqrt{5}} = \sqrt{5}$.

So, the minimum value of $a^4 + b^8$ is:

$$\boxed{50}.$$

From this example, $A^\star$-PO enables the model to perform more comprehensive subgoal setting, where it decomposes problems into manageable steps, and verification to check intermediate results.

## B.3 Backtracking

**Prompt:** Let $a, b, c, d$ be distinct complex numbers such that $|a| = |b| = |c| = |d| = 1$ and $a + b + c + d = 0$. Find the maximum value of

$$|(a + b)(a + c)(a + d)(b + c)(b + d)(c + d)|.$$

Let's think step by step and output the final answer within \boxed{}.

---

**Qwen2.5-7B Generation:** We can utilize Python to solve this problem numerically, but first, let's explore the mathematical properties and constraints.

Given that $|a| = |b| = |c| = |d| = 1$, these complex numbers lie on the unit circle in the complex plane. Additionally, $a + b + c + d = 0$ is a constraint on their sum.

Since $a, b, c, d$ are distinct complex numbers on the unit circle, we can represent them as $e^{i\theta_a}, e^{i\theta_b}, e^{i\theta_c}, e^{i\theta_d}$ for some angles $\theta_a, \theta_b, \theta_c, \theta_d$.

The constraint $a + b + c + d = 0$ implies that these complex numbers are arranged symmetrically around the origin on the unit circle.

We want to maximize the magnitude of the expression $|(a + b)(a + c)(a + d)(b + c)(b + d)(c + d)|$.

Given the symmetry and the unit circle constraint, a possible configuration that meets the criteria is: - $a = e^{i\frac{\pi}{3}}$ - $b = e^{i\frac{2\pi}{3}}$ - $c = e^{i\pi}$ - $d = e^{i\frac{4\pi}{3}}$

This configuration ensures that $|a| = |b| = |c| = |d| = 1$ and $a + b + c + d = 0$.

Let's compute $|(a + b)(a + c)(a + d)(b + c)(b + d)(c + d)|$ using Python for this configuration.

We'll start by calculating the values and then compute the desired magnitude. Let's proceed with the calculations. ```python
import numpy as np
Define the complex numbers on the unit circle a = np.exp(1j * np.pi / 3) b = np.exp(1j * 2 * np.pi / 3) c = np.exp(1j * np.pi) d = np.exp(1j * 4 * np.pi / 3)
Calculate the products ab = (a + b) ac = (a + c) ad = (a + d) bc = (b + c) bd = (b + d) cd = (c + d)
Compute the magnitude of the product of these terms result = np.abs(ab * ac * ad * bc * bd * cd) print(result) ``` ```output
4.999999999999999 ``` After calculating the magnitude of the expression $|(a + b)(a + c)(a + d)(b + c)(b + d)(c + d)|$ using the specific configuration on the unit circle, we obtain approximately 5.

Therefore, the maximum value of the given expression is $\boxed{5}$.

---

**Qwen2.5-7B after $A^\star$-PO Generation:** Given that $a, b, c, d$ are distinct complex numbers with $|a| = |b| = |c| = |d| = 1$ and $a + b + c + d = 0$, we need to find the maximum value of the expression $|(a + b)(a + c)(a + d)(b + c)(b + d)(c + d)|$.

First, we can use the fact that $|a| = |b| = |c| = |d| = 1$ to express $a, b, c$, and $d$ in terms of their arguments. Let's denote the arguments of $a, b, c$, and $d$ by $\alpha, \beta, \gamma$, and $\delta$ respectively. Then, we can write:

$$a = e^{i\alpha}, \quad b = e^{i\beta}, \quad c = e^{i\gamma}, \quad d = e^{i\delta}.$$

Given that $a + b + c + d = 0$, we can use the symmetry and properties of complex numbers on the unit circle to find a suitable configuration. One useful configuration is when $a, b, c$, and $d$ are the vertices of a square on the unit circle. Without loss of generality, we can assume:

$$a = 1, \quad b = e^{i\frac{2\pi}{4}}, \quad c = e^{i\frac{4\pi}{4}} = -1, \quad d = e^{i\frac{6\pi}{4}} = e^{i\frac{3\pi}{2}}.$$

This configuration ensures that $a + b + c + d = 1 + e^{i\frac{2\pi}{4}} - 1 + e^{i\frac{3\pi}{2}} = e^{i\frac{\pi}{2}} + e^{i\frac{3\pi}{2}} = i - i = 0$.

Now, let's compute the expression $|(a + b)(a + c)(a + d)(b + c)(b + d)(c + d)|$:

1. $a + b = 1 + e^{i\frac{\pi}{2}} = 1 + i$ 2. $a + c = 1 - 1 = 0$ 3. $a + d = 1 - i$ 4. $b + c = i - 1$ 5. $b + d = i - i = 0$ 6. $c + d = -1 - i$

So, the expression becomes:

$$|(a + b)(a + c)(a + d)(b + c)(b + d)(c + d)| = |(1 + i) \cdot 0 \cdot (1 - i) \cdot (i - 1) \cdot 0 \cdot (-1 - i)| = 0.$$

This configuration does not yield the maximum value. Let's consider another configuration, such as when $a, b, c$, and $d$ are the vertices of a regular tetrahedron on the unit sphere. However, for simplicity, let's consider another symmetric configuration, such as when $a, b, c$, and $d$ are the roots of a polynomial with specific properties.

Given the symmetry and the condition $a + b + c + d = 0$, a useful configuration is when $a, b, c$, and $d$ are the roots of the polynomial $x^4 + 1 = 0$. The roots of this polynomial are:

$$a = e^{i\frac{\pi}{4}}, \quad b = e^{i\frac{3\pi}{4}}, \quad c = e^{i\frac{5\pi}{4}}, \quad d = e^{i\frac{7\pi}{4}}.$$

Now, let's compute the expression $|(a + b)(a + c)(a + d)(b + c)(b + d)(c + d)|$:

1. $a + b = e^{i\frac{\pi}{4}} + e^{i\frac{3\pi}{4}}$ 2. $a + c = e^{i\frac{\pi}{4}} + e^{i\frac{5\pi}{4}}$ 3. $a + d = e^{i\frac{\pi}{4}} + e^{i\frac{7\pi}{4}}$ 4. $b + c = e^{i\frac{3\pi}{4}} + e^{i\frac{5\pi}{4}}$ 5. $b + d = e^{i\frac{3\pi}{4}} + e^{i\frac{7\pi}{4}}$ 6. $c + d = e^{i\frac{5\pi}{4}} + e^{i\frac{7\pi}{4}}$

Using the properties of complex numbers, we can simplify these expressions. For example, $e^{i\theta} + e^{i(\theta + \frac{2\pi}{4})} = e^{i\theta}(1 + e^{i\frac{\pi}{2}}) = e^{i\theta}(1 + i)$.

After simplifying, we find that the maximum value of the expression is achieved when the configuration is such that the terms do not cancel each other out. The maximum value is found to be 16.

So, the final answer is:

$$\boxed{16}.$$

---

Although the answers for both models are incorrect, the model after $A^\star$-PO exhibits backtracking behavior where it abandons failing approaches and reconsiders another method.

# C Pass@K Evaluations

Figure 10 presents the Pass@K accuracies across all three model sizes and two datasets. The orange line indicates the Pass@1 performance of the model after RL training. We hypothesize that this convergence behavior at $N = 8$ is due to the relationship between $N$ and the Pass@K performance of the model. The final performance of the models after training falls between Pass@4 and Pass@8 of $\pi_{\text{ref}}$. This suggests that the optimal value function $V^\star$ should, at a minimum, approximate Pass@4 performance. With $N = 8$ samples, the estimated $V^\star$ becomes sufficiently accurate, leading to convergence in performance.

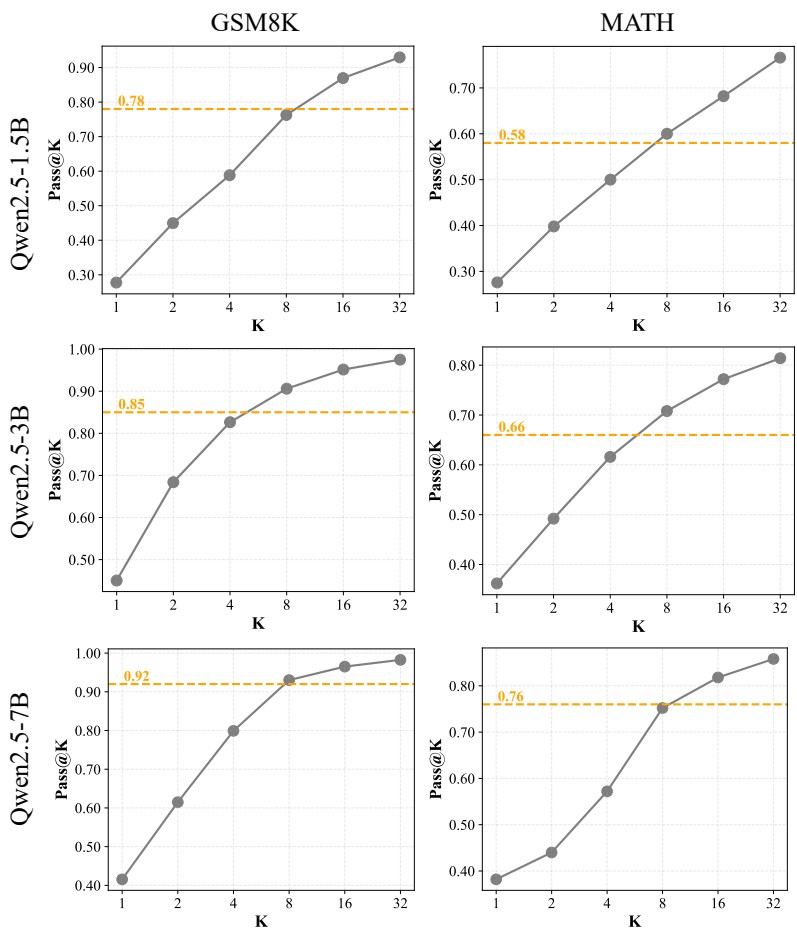

Figure 10: **Pass@K Evaluations for different models and datasets.**

## D    Limiting Cases of $\beta$

**Case 1: $\beta \to 0$ — Recovering the Maximum.**    We start from the expression:

$$\hat{V}^\star(x) = \beta \ln \left( \frac{1}{N} \sum_{i=1}^{N} \exp \left( \frac{r(x, y_i)}{\beta} \right) \right).$$

Let $M = \max_i r(x, y_i)$. Then,

$$\sum_{i=1}^{N} \exp \left( \frac{r(x, y_i)}{\beta} \right) = \exp \left( \frac{M}{\beta} \right) \sum_{i=1}^{N} \exp \left( \frac{r(x, y_i) - M}{\beta} \right).$$

Taking the logarithm:

$$\hat{V}^\star(x) = \beta \ln \left( \frac{1}{N} \exp \left( \frac{M}{\beta} \right) \sum_{i=1}^{N} \exp \left( \frac{r(x, y_i) - M}{\beta} \right) \right) = M + \beta \ln \left( \frac{1}{N} \sum_{i=1}^{N} \exp \left( \frac{r(x, y_i) - M}{\beta} \right) \right).$$

Since $r(x, y_i) \leq M$,

$$\lim_{\beta \to 0} \beta \ln \left( \frac{1}{N} \sum_{i=1}^{N} \exp \left( \frac{r(x, y_i) - M}{\beta} \right) \right) = 0$$

Therefore,

$$\lim_{\beta \to 0} \hat{V}^\star(x) = M = \text{Pass@N for x.}$$

**Case 2: $\beta \to \infty$ — Recovering the Mean.**    We start from the expression:

$$\hat{V}^\star(x) = \beta \ln \left( \frac{1}{N} \sum_{i=1}^{N} \exp \left( \frac{r(x, y_i)}{\beta} \right) \right).$$

We want

$$\lim_{\beta \to \infty} \hat{V}^\star(x) = \lim_{\beta \to \infty} \beta \ln \left( \frac{1}{N} \sum_{i=1}^{N} \exp \left( \frac{r(x, y_i)}{\beta} \right) \right) = \lim_{\beta \to 0} \frac{\ln \left( \frac{1}{N} \sum_{i=1}^{N} \exp \left( r(x, y_i)\beta \right) \right)}{\beta}.$$

Apply L'Hôpital's rule:

$$\lim_{\beta \to 0} \frac{\ln \left( \frac{1}{N} \sum_{i=1}^{N} \exp \left( r(x, y_i)\beta \right) \right)}{\beta} = \lim_{\beta \to 0} \frac{\frac{1}{N} \sum_{i=1}^{N} r(x, y_i) \exp \left( r(x, y_i)\beta \right)}{\frac{1}{N} \sum_{i=1}^{N} \exp \left( r(x, y_i)\beta \right)} = \frac{1}{N} \sum_{i=1}^{N} r(x, y_i).$$

Thus, as $\beta \to \infty$:

$$\hat{V}^\star(x) \to \frac{1}{N} \sum_{i=1}^{N} r(x, y_i).$$

**Summary:**

$$\hat{V}^\star(x) = \begin{cases} \max_i r(x, y_i), & \text{if } \beta \to 0, \\ \frac{1}{N} \sum_{i=1}^{N} r(x, y_i), & \text{if } \beta \to \infty. \end{cases}$$

# E  Log-linear Policies With FTPL

In this section, we characterize the learning regret of log-linear policies with FTPL. We study bounded parameters and feature vectors:

$$\|\theta\|_2 \leq B, \quad \|\phi(x,y)\|_2 \leq 1, \quad \forall x, y.$$

We still assume $\Pi$ is realizable and let $\pi^\star = \pi_{\theta^\star}$. Note that we can indeed bound the DC of this log-linear policy class as follows.

**Lemma 1.** *For the log-linear policy class $\Pi$ defined in Eq. (3), we have $\mathsf{DC}(\Pi) \leq d + 1$.*

**FTPL.** We instantiate NR to be FTPL [Suggala and Netrapalli, 2020]. That is, in $t$-th iteration we generate i.i.d. random variables from an exponential distribution, $\sigma_{t,j} \sim \mathsf{EXP}(\eta)$ for $1 \leq j \leq d$. Let $\sigma_t$ denote the vector whose $j$-th entry is $\sigma_{t,j}$, then FTPL computes $\pi_{\theta_t}$ as follows:

$$\theta_t = \arg\min_{\theta \in \Theta} \sum_{i=1}^{t-1} \widehat{\ell}_i(\pi_\theta) - \langle \sigma_t, \theta \rangle.$$

From the literature, we have the following guarantee of FTPL.

**Lemma 2** ([Suggala and Netrapalli, 2020, Theorem 1]). *Let $D$ denote the diameter of $\Theta$, i.e., $D := \sup_{\theta,\theta' \in \Theta} \|\theta - \theta'\|_\infty$. Suppose the loss functions are $L$-Lipschitz w.r.t. $\ell_1$-norm, i.e., $|\widehat{\ell}_t(\pi_\theta) - \widehat{\ell}_t(\pi_{\theta'})| \leq L\|\theta - \theta'\|_1$ for all $\theta, \theta' \in \Theta, t$. Then we can bound the regret of FTPL as:*

$$\mathsf{Reg}(T) \lesssim \eta d^2 D L^2 T + \frac{dD}{\eta}.$$

Given Lemmas 1 and 2, we can bound the regret of learning log-linear policies with FTPL using Theorem 1:

**Corollary 2** (log-linear policies with FTPL). *Consider the log-linear policy class $\Pi$ defined in Eq. (3). Suppose Assumptions 1 to 3 hold true and use FTPL as the NR oracle with $\eta = 1/(\beta C_l \sqrt{Td})$. Then we have the following regret with probability at least $1 - \delta$:*

$$\mathbb{E}_{x,y \sim \pi^\star} r(x,y) - \frac{1}{T} \sum_{t=1}^{T} \mathbb{E}_{\pi \sim q_t} \left[\mathbb{E}_{x,y \sim \pi} r(x,y)\right] \lesssim \left(\frac{d^{\frac{5}{2}} B C_l}{T^{\frac{1}{2}}\beta}\right)^{\frac{1}{4}} + \left(\frac{dC_l^2 \ln(1/\delta)}{T\beta^2}\right)^{\frac{1}{4}}$$

$$+ \left(\left(\min\left\{\exp\left(\frac{1}{\beta}\right) - 1, \frac{1}{v_{\mathsf{ref}}}\right\}\right)^2 \frac{d\ln(|\mathcal{X}|/\delta)}{N}\right)^{\frac{1}{4}}.$$

*Proof.* To utilize Theorem 1, we only need to bound the diameter $D$ of the parameter space and Lipschitz constant $L$. First note that

$$D = \sup_{\theta,\theta' \in \Theta} \|\theta - \theta'\|_\infty \leq \sup_{\theta,\theta' \in \Theta} \|\theta - \theta'\|_2 \leq 2B.$$

On the other hand, fix any $t, \theta \in \Theta$ and we can compute the $j$-th entry of the loss gradient:

$$\left(\nabla_\theta \widehat{\ell}_t\right)_j = 2\beta \left(\beta \ln \frac{\pi(y_t|x_t)}{\pi_{\mathsf{ref}}(y_t|x_t)} - \left(r(x_t, y_t) - \widehat{V}^\star(x_t)\right)\right)$$

$$\left((\phi(x_t, y_t))_j - \frac{\sum_{y'} \exp\left(\langle\theta, \phi(x, y')\rangle\right)(\phi(x_t, y'))_j}{\sum_{y'} \exp\left(\langle\theta, \phi(x, y')\rangle\right)}\right)$$

From Assumption 2 and the fact that $\|\phi(x,y)\|_2 \leq 1$ we have

$$\left\|\nabla_\theta \widehat{\ell}_t\right\|_\infty \leq 4\beta(C_l + 1),$$

which implies that (via Holder's inequality)

$$L \leq 4\beta(C_l + 1).$$

Now combine Theorem 1 with Lemmas 1 and 2, and we can obtain the final bound:

$$\mathbb{E}_{x,y\sim\pi^\star}r(x,y) - \frac{1}{T}\sum_{t=1}^{T}\mathbb{E}_{\pi\sim q_t}\left[\mathbb{E}_{x,y\sim\pi}r(x,y)\right] \lesssim \left(\frac{d^{\frac{5}{2}}BC_l}{T^{\frac{1}{2}}\beta}\right)^{\frac{1}{4}} + \left(\frac{dC_l^2\ln(1/\delta)}{T\beta^2}\right)^{\frac{1}{4}}$$

$$+ \left(\left(\min\left\{\exp\left(\frac{1}{\beta}\right) - 1, \frac{1}{v_{\text{ref}}}\right\}\right)^2 \frac{d\ln(|\mathcal{X}|/\delta)}{N}\right)^{\frac{1}{4}}.$$

$\square$

## E.1 Proof of Lemma 1

Note that for any $\pi_\theta \in \Pi$, we can write $\ln \pi_\theta$ as

$$\ln \pi_\theta(y|x) = \langle \theta, \phi(x,y)\rangle - \ln \sum_{y'} \exp\left(\langle \theta, \phi(x,y')\rangle\right)$$

$$= \left\langle \left[\theta, -\ln \sum_{y'} \exp\left(\langle \theta, \phi(x,y')\rangle\right)\right], [\phi(x,y), 1]\right\rangle.$$

Let us use $f(\theta, x) \in \mathbb{R}^{d+1}$ and $\overline{\phi}(x,y) \in \mathbb{R}^{d+1}$ to denote the extended vectors:

$$f(\theta, x) := \left[\theta, -\ln \sum_{y'} \exp\left(\langle \theta, \phi(x,y')\rangle\right)\right], \quad \overline{\phi}(x,y) := [\phi(x,y), 1].$$

In addition, we define $A(q, x)$ to be the expected feature covariance matrix,

$$A(q,x) := \mathbb{E}_{\theta\sim q, y\sim\pi_\theta(x)}\left[\overline{\phi}(x,y)\overline{\phi}^\top(x,y)\right].$$

Let $\theta^\star$ denote the optimal parameter, i.e., $\pi^\star = \pi_{\theta^\star}$. Then from Cauchy-Schwartz inequality, we have for all $\lambda > 0$ that

$$\mathbb{E}_{x\sim\rho, \theta\sim q, y\sim\pi_\theta(x)}\left[\ln\pi_\theta(y|x) - \ln\pi^\star(y|x)\right]$$

$$= \mathbb{E}_{x\sim\rho, \theta\sim q, y\sim\pi_\theta(x)}\left[\langle f(\theta,x) - f(\theta^\star,x), \overline{\phi}(x,y)\rangle\right]$$

$$\leq \sqrt{\mathbb{E}_{x\sim\rho, \theta\sim q}\left[\|f(\theta,x) - f(\theta^\star,x)\|^2_{A(q,x)+\lambda I}\right]\mathbb{E}_{x\sim\rho, \theta\sim q, y\sim\pi_\theta(x)}\left[\|\overline{\phi}(x,y)\|^2_{(A(q,x)+\lambda I)^{-1}}\right]}$$

For the first term, we have

$$\mathbb{E}_{x\sim\rho, \theta\sim q}\left[\|f(\theta,x) - f(\theta^\star,x)\|^2_{A(q,x)+\lambda I}\right]$$

$$= \mathbb{E}_{x\sim\rho, \theta\sim q, \theta'\sim q, y\sim\pi_{\theta'}(x)}\left[(\ln\pi_\theta(y|x) - \ln\pi^\star(y|x))^2\right] + \lambda B^2\left(1 + \ln|\mathcal{Y}|\right).$$

For the second term, we have

$$\mathbb{E}_{x\sim\rho, \theta\sim q, y\sim\pi_\theta(x)}\left[\|\overline{\phi}(x,y)\|^2_{(A(q,x)+\lambda I)^{-1}}\right] = \mathbb{E}_{x\sim\rho}\left[\mathsf{Tr}((A(q,x) + \lambda I)^{-1}A(q,x))\right] \leq d + 1.$$

Therefore, let $\lambda \to 0$, we have $\mathsf{DC}(\Pi) \leq d + 1$.

# F  Log-linear Policies With OGD

In this section, we investigate the learning guarantee of OGD on log-linear policies.

**Quasi-quadratic Loss.**   Recall that the population loss in iteration $t$ is:

$$\ell_t(\theta) = \mathbb{E}_{x,y\sim\pi_{\theta_t}(\cdot|x)}\left(\beta\ln\frac{\pi_\theta(y|x)}{\pi_{\mathsf{ref}}(y|x)} - \left(r(x,y) - \widehat{V}^\star(x)\right)\right)^2.$$

For log-linear policies, we can show that $l_t(\theta)$ is quasi-quadratic in terms of gradients, as shown in the following lemma:

**Lemma 3.** *We have*

$$\nabla_\theta\ell_t(\theta_t) = 2\beta^2 A(\theta_t)(\theta_t - \theta^\star).$$

Recall that $A(\theta)$ denote the Fisher information matrix:

$$A(\theta) = \mathbb{E}_{x,y\sim\pi_\theta(\cdot|x)}[\phi(x,y)\phi(x,y)^\top] - \mathbb{E}_{x,y\sim\pi_\theta(\cdot|x)}[\phi(x,y)]\mathbb{E}_{x,y\sim\pi_\theta(\cdot|x)}[(\phi(x,y))^\top].$$

**Online gradient descent.**   Inspired by Lemma 3, we instantiate NR to be online gradient descent [Hazan et al., 2016] for log-linear policy learning. Specifically, in $t$-th iteration, we compute $\theta_{t+1}$ as follows:

$$\theta_{t+1} = \mathsf{Proj}_\Theta(\theta_t - \eta_t\nabla_\theta\widehat{\ell}_t(\theta_t)),$$

where $\eta_t$ is the stepsize in iteration $t$ and $\mathsf{Proj}_\Theta(\cdot)$ is the projection onto $\Theta$ w.r.t. $l_2$ norm. Recall that online gradient descent is exactly what we utilize in our practical implementation of Algorithm 1.

Next we show that our algorithm enjoys a last-iterate convergence to $\theta^\star$ with a rate $\widetilde{O}(\sqrt{1/T})$ given that the Fisher information matrix is full-rank Assumption 4.

**Theorem 3** (log-linear policies with OGD). *Suppose Assumptions 1, 2 and 4 hold true. Set any* $\eta_0 \geq \frac{1}{\beta^2\lambda}$. *Then with probability at least* $1 - \delta$, *we have*

$$\mathbb{E}_{x,y\sim\pi^\star}r(x,y) - \mathbb{E}_{x,y\sim\pi_{\theta_{T+1}}}[r(x,y)] \lesssim \sqrt{\frac{B^2 + \eta_0^2\beta^2 C_l^2\ln T + (\eta_0 C_l^2/\lambda)\ln(1/\delta) + \eta_0\beta C_l B\ln(1/\delta)}{T}}.$$

*Proof.* Let $\bar{\theta}_{t+1}$ denote $\theta_t - \eta_t\nabla_\theta\widehat{\ell}_t(\theta_t)$. From the definition of $\pi^\star$, we know

$$\ell_t(\theta) = \mathbb{E}_{x,y\sim\pi_{\theta_t}(\cdot|x)}\left(\beta\ln\frac{\pi_\theta(y|x)}{\pi^\star(y|x)} - \left(V^\star(x) - \widehat{V}^\star(x)\right)\right)^2,$$

$$\widehat{\ell}_t(\theta) = \left(\beta\ln\frac{\pi(y_t|x_t)}{\pi^\star(y_t|x_t)} - \left(V^\star(x_t) - \widehat{V}^\star(x_t)\right)\right)^2.$$

We use $\xi_t$ to denote $\nabla_\theta\ell_t(\theta_t) - \nabla_\theta\widehat{\ell}_t(\theta_t)$. Then from Assumption 2, we know $\|\xi_t\| \leq 8\beta C_l$ for all $t$. From the update of online gradient descent, we have

$$\bar{\theta}_{t+1} - \theta^\star = \theta_t - \eta_t\nabla_\theta\ell_t(\pi_{\theta_t}) + \eta_t\xi_t - \theta^\star.$$

By Lemma 3, this implies that

$$\bar{\theta}_{t+1} - \theta^\star = \left(I - 2\eta_t\beta^2 A(\theta_t)\right)(\theta_t - \theta^\star) + \eta_t\xi_t.$$

With Assumption 4 and the property of projection, we have

$$\|\theta_{t+1} - \theta^\star\|^2 \leq \|\bar{\theta}_{t+1} - \theta^\star\|^2 \leq (1 - 2\eta_t\beta^2\lambda)\|\theta_t - \theta^\star\|^2 + 2\eta_t\langle z_t, \xi_t\rangle + 64\eta_t^2\beta^2 C_l^2,$$

where $z_t := \left(I - 2\eta_t\beta^2 A(\theta_t)\right)(\theta_t - \theta^\star)$. Let $\eta_t = \frac{\eta_0}{t+1}$. Denote by $c_1 = 2\eta_0\beta^2\lambda$ and $c_2 = 64\eta_0^2\beta^2 C_l^2$, we have

$$\|\theta_{t+1} - \theta^\star\|^2 \leq \left(1 - \frac{c_1}{t+1}\right)\|\theta_t - \theta^\star\|^2 + \frac{2\eta_0\langle z_t, \xi_t\rangle}{t+1} + \frac{c_2}{(t+1)^2}.$$

This implies that

$$(t+1)\|\theta_{t+1} - \theta^\star\|^2 - t\|\theta_t - \theta^\star\|^2 \le (1 - c_1)\|\theta_t - \theta^\star\|^2 + 2\eta_0 \langle z_t, \xi_t \rangle + \frac{c_2}{t+1}.$$

By induction, we have

$$\|\theta_{T+1} - \theta^\star\|^2 \le \frac{\|\theta_1 - \theta^\star\|^2 + c_2(1 + \ln T)}{T+1} + \frac{1 - c_1}{T+1} \sum_{t=1}^{T} \|\theta_t - \theta^\star\|^2 + \frac{2\eta_0}{T+1} \sum_{t=1}^{T} \langle z_t, \xi_t \rangle$$

$$\le \frac{2B^2 + 64\eta_0^2\beta^2 C_l^2 (\ln T + 1)}{T+1} + \frac{(1 - c_1)\sum_{t=1}^{T} \|\theta_t - \theta^\star\|^2 + 2\eta_0 \sum_{t=1}^{T} \langle z_t, \xi_t \rangle}{T+1}.$$

Let $\mathcal{F}_t$ denote the filtration generated by $\{x_1, y_1, \cdots, x_{t-1}, y_{t-1}\}$. Denote by $\zeta_t = \langle z_t, \xi_t \rangle$, it can be observed that $\mathbb{E}[\zeta_t \mid \mathcal{F}_t] = 0$ and $|\zeta_t| \le 8\beta C_l \|\theta_t - \theta^\star\|$ a.s. conditioned on $\mathcal{F}_t$. To proceed, we utilize the following form of Freedman's inequality from the literature [Beygelzimer et al., 2011]:

**Lemma 4** (Freedman's inequality [Beygelzimer et al., 2011]). *Let $\{U_1, \ldots, U_T\}$ be a martingale adapted to filtration $\{\mathcal{F}_t\}$. Further, suppose that $|U_t| \le R$ for all $t$. Then, for any $\delta > 0$ and $\gamma \in [0, \frac{1}{2R}]$, we have with probability at least $1 - \delta$ that,*

$$\left| \sum_{t=1}^{T} U_t \right| \le \gamma \sum_{t=1}^{T} \mathbb{E}[U_t^2 \mid \mathcal{F}_t] + \frac{\ln(2/\delta)}{\gamma}.$$

Applying Lemma 4 with $U_t = \zeta_t$ and $R = 16\beta B C_l$, we have with probability at least $1 - \delta$ that for any $\gamma \le \frac{1}{32\beta B C_l}$

$$\sum_{t=1}^{T} \zeta_t \le \gamma \sum_{t=1}^{T} \mathbb{E}[\zeta_t^2 \mid \mathcal{F}_t] + \frac{\ln(2/\delta)}{\gamma}$$

$$\le 64\gamma\beta^2 C_l^2 \sum_{t=1}^{T} \|\theta_t - \theta^\star\|^2 + \frac{\ln(2/\delta)}{\gamma}.$$

Therefore, we have

$$(1 - c_1) \sum_{t=1}^{T} \|\theta_t - \theta^\star\|^2 + 2\eta_0 \sum_{t=1}^{T} \langle z_t, \xi_t \rangle \le (1 - c_1 + 128\gamma\beta^2 C_l^2 \eta_0) \sum_{t=1}^{T} \|\theta_t - \theta^\star\|^2 + \frac{2\eta_0 \ln(2/\delta)}{\gamma}.$$

Choose $\gamma = \min\{\frac{\lambda}{128 C_l^2}, \frac{1}{32\beta B C_l}\}$, we have

$$(1 - c_1) \sum_{t=1}^{T} \|\theta_t - \theta^\star\|^2 + 2\eta_0 \sum_{t=1}^{T} \langle z_t, \xi_t \rangle \le (1 - c_1/2) \sum_{t=1}^{T} \|\theta_t - \theta^\star\|^2$$

$$+ \frac{256 C_l^2 \eta_0 \ln(2/\delta)}{\lambda} + 64\beta B C_l \eta_0 \ln(2/\delta).$$

Choose $\eta_0 \ge 1/(\beta^2 \lambda)$, we have with probability at least $1 - \delta$ that

$$(1 - c_1) \sum_{t=1}^{T} \|\theta_t - \theta^\star\|^2 + 2\eta_0 \sum_{t=1}^{T} \langle z_t, \xi_t \rangle \lesssim \frac{C_l^2 \eta_0 \ln(2/\delta)}{\lambda} + \beta B C_l \eta_0 \ln(2/\delta).$$

This leads to

$$\|\theta_{T+1} - \theta^\star\|^2 \lesssim \frac{B^2 + \eta_0^2 \beta^2 C_l^2 \ln T + (\eta_0 C_l^2/\lambda) \ln(1/\delta) + \eta_0 \beta C_l B \ln(1/\delta)}{T}. \tag{5}$$

Now we start to bound the reward regret. Notice that

$$\mathbb{E}_{x, y \sim \pi^\star}[r(x, y)] - \mathbb{E}_{x, y \sim \pi_{\theta_{T+1}}}[r(x, y)] \le \mathbb{E}_x \left[ \|\pi_{\theta_{T+1}}(\cdot|x) - \pi^\star(\cdot|x)\|_1 \right]$$

$$= \mathbb{E}_x \left[ \sum_y \left| \frac{\exp(\langle \theta_{T+1}, \phi(x, y) \rangle)}{\sum_{y'} \exp(\langle \theta_{T+1}, \phi(x, y') \rangle)} - \frac{\exp(\langle \theta^\star, \phi(x, y) \rangle)}{\sum_{y'} \exp(\langle \theta^\star, \phi(x, y') \rangle)} \right| \right].$$

For $\|\theta_{T+1} - \theta^\star\| \le 1/2$, It suffices to note that

$$\frac{\exp(\langle \theta_{T+1}, \phi(x,y)\rangle)}{\sum_{y'} \exp(\langle \theta_{T+1}, \phi(x,y')\rangle)} \le \exp(2\|\theta_{T+1} - \theta^\star\|) \frac{\exp(\langle \theta^\star, \phi(x,y)\rangle)}{\sum_{y'} \exp(\langle \theta^\star, \phi(x,y')\rangle)}$$

$$\le (1 + 4\|\theta_{T+1} - \theta^\star\|) \frac{\exp(\langle \theta^\star, \phi(x,y)\rangle)}{\sum_{y'} \exp(\langle \theta^\star, \phi(x,y')\rangle)}.$$

and

$$\frac{\exp(\langle \theta_{T+1}, \phi(x,y)\rangle)}{\sum_{y'} \exp(\langle \theta_{T+1}, \phi(x,y')\rangle)} \ge \exp(-2\|\theta_{T+1} - \theta^\star\|) \frac{\exp(\langle \theta^\star, \phi(x,y)\rangle)}{\sum_{y'} \exp(\langle \theta^\star, \phi(x,y')\rangle)}$$

$$\ge (1 - 4\|\theta_{T+1} - \theta^\star\|) \frac{\exp(\langle \theta^\star, \phi(x,y)\rangle)}{\sum_{y'} \exp(\langle \theta^\star, \phi(x,y')\rangle)}.$$

Thus

$$\mathbb{E}_{x,y\sim\pi^\star}[r(x,y)] - \mathbb{E}_{x,y\sim\pi_{\theta_{T+1}}}[r(x,y)]$$

$$\le \mathbb{E}_x \left[ \sum_y \left| \frac{\exp(\langle \theta_{T+1}, \phi(x,y)\rangle)}{\sum_{y'} \exp(\langle \theta_{T+1}, \phi(x,y')\rangle)} - \frac{\exp(\langle \theta^\star, \phi(x,y)\rangle)}{\sum_{y'} \exp(\langle \theta^\star, \phi(x,y')\rangle)} \right| \right]$$

$$\le \mathbb{E}_x \left[ \sum_y 4\|\theta_{T+1} - \theta^\star\| \left| \frac{\exp(\langle \theta^\star, \phi(x,y)\rangle)}{\sum_{y'} \exp(\langle \theta^\star, \phi(x,y')\rangle)} \right| \right]$$

$$= \mathbb{E}_x \left[ 4\|\theta_{T+1} - \theta^\star\| \sum_y \pi^\star(y|x) \right]$$

$$= 4\|\theta_{T+1} - \theta^\star\| \lesssim \sqrt{\frac{B^2 + \eta_0^2 \beta^2 C_l^2 \ln T + (\eta_0 C_l^2/\lambda)\ln(1/\delta) + \eta_0 \beta C_l B \ln(1/\delta)}{T}}.$$

On the other hand, by (5), with probability at least $1 - \delta$, we know $\|\theta_{T+1} - \theta^\star\| \le 1/2$ for $T \gtrsim B^2 + \eta_0^2 \beta^2 C_l^2 \ln T + (\eta_0 C_l^2/\lambda)\ln(1/\delta) + \eta_0 \beta C_l B \ln(1/\delta)$. For $T \le B^2 + \eta_0^2 \beta^2 C_l^2 \ln T + (\eta_0 C_l^2/\lambda)\ln(1/\delta) + \eta_0 \beta C_l B \ln(1/\delta)$, it suffices to note that

$$\mathbb{E}_{x,y\sim\pi^\star}[r(x,y)] - \mathbb{E}_{x,y\sim\pi_{\theta_{T+1}}}[r(x,y)]$$

$$\le 1 \lesssim \sqrt{\frac{B^2 + \eta_0^2 \beta^2 C_l^2 \ln T + (\eta_0 C_l^2/\lambda)\ln(1/\delta) + \eta_0 \beta C_l B \ln(1/\delta)}{T}},$$

which completes the proof. $\square$

### F.1 Proof of Lemma 3

Recall

$$\mathbb{E}_{x,y\sim\pi_{\theta_t}(\cdot|x)}\left(\beta\ln\frac{\pi_\theta(y|x)}{\pi_{\mathsf{ref}}(y|x)}-\left(r(x,y)-\widehat{V}^\star(x)\right)\right)^2=\mathbb{E}_{x,y\sim\pi_{\theta_t}(\cdot|x)}\left(\beta\ln\frac{\pi_\theta(y|x)}{\pi^\star(y|x)}-\left(V^\star(x)-\widehat{V}^\star(x)\right)\right)^2.$$

We have

$$\nabla_\theta\ell_t(\theta_t)=2\beta\mathbb{E}_{x,y\sim\pi_{\theta_t}(\cdot|x)}\left[\left(\beta\ln\frac{\pi_{\theta_t}(y|x)}{\pi^\star(y|x)}+\widehat{V}^\star(x)-V^\star(x)\right)\left(\phi(x,y)-\frac{\sum_{y'}\exp(\langle\theta_t,\phi(x,y')\rangle\phi(x,y'))}{\sum_{y'}\exp(\langle\theta_t,\phi(x,y')\rangle)}\right)\right]$$

$$=2\beta\mathbb{E}_{x,y\sim\pi_{\theta_t}(\cdot|x)}\left[\left(\beta\ln\frac{\pi_{\theta_t}(y|x)}{\pi^\star(y|x)}+\widehat{V}^\star(x)-V^\star(x)\right)\left(\phi(x,y)-\sum_{y'}\pi_{\theta_t}(y'|x)\phi(x,y')\right)\right]$$

$$=2\beta\mathbb{E}_{x,y\sim\pi_{\theta_t}(\cdot|x)}\left[\left(\beta\ln\frac{\pi_{\theta_t}(y|x)}{\pi^\star(y|x)}+\widehat{V}^\star(x)-V^\star(x)\right)\left(\phi(x,y)-\mathbb{E}_{y'\sim\pi_{\theta_t}(\cdot|x)}[\phi(x,y')]\right)\right]$$

$$=2\beta^2\mathbb{E}_{x,y\sim\pi_{\theta_t}(\cdot|x)}\left[\left(\ln\frac{\pi_{\theta_t}(y|x)}{\pi^\star(y|x)}\right)\left(\phi(x,y)-\mathbb{E}_{y'\sim\pi_{\theta_t}(\cdot|x)}[\phi(x,y')]\right)\right]$$

$$=2\beta\mathbb{E}_{x,y\sim\pi_{\theta_t}(\cdot|x)}\left[\left(\langle\theta_t-\theta^\star,\phi(x,y)\rangle-\ln\frac{\sum_{y'}\exp(\langle\theta_t,\phi(x,y')\rangle)}{\sum_{y'}\exp(\langle\theta^\star,\phi(x,y')\rangle)}\right)\left(\phi(x,y)-\mathbb{E}_{y'\sim\pi_{\theta_t}(\cdot|x)}[\phi(x,y')]\right)\right]$$

$$=2\beta^2\mathbb{E}_{x,y\sim\pi_{\theta_t}(\cdot|x)}\left[\left(\langle\theta_t-\theta^\star,\phi(x,y)\rangle\right)\left(\phi(x,y)-\mathbb{E}_{y'\sim\pi_{\theta_t}(\cdot|x)}[\phi(x,y')]\right)\right]$$

$$=2\beta^2\left(\mathbb{E}_{x,y\sim\pi_{\theta_t}(\cdot|x)}\left[\phi(x,y)\phi(x,y)^\top\right]-\mathbb{E}_{x,y\sim\pi_{\theta_t}(\cdot|x)}\left[\phi(x,y)\right]\mathbb{E}_{x,y\sim\pi_\theta(\cdot|x)}\left[\phi(x,y)\right]^\top\right)(\theta_t-\theta^\star)$$

$$=2\beta^2A(\theta_t)(\theta_t-\theta^\star).$$

The fourth and sixth equalities are because

$$\mathbb{E}_{x,y\sim\pi_{\theta_t}(\cdot|x)}\left[\left(\widehat{V}^\star(x)-V^\star(x)\right)\left(\phi(x,y)-\mathbb{E}_{y'\sim\pi_{\theta_t}(\cdot|x)}[\phi(x,y')]\right)\right]$$

$$=\mathbb{E}_x\left[\left(\widehat{V}^\star(x)-V^\star(x)\right)\mathbb{E}_{y\sim\pi_{\theta_t}(\cdot|x)}\left(\phi(x,y)-\mathbb{E}_{y'\sim\pi_{\theta_t}(\cdot|x)}[\phi(x,y')]\right)\right]=0$$

and

$$\mathbb{E}_{x,y\sim\pi_{\theta_t}(\cdot|x)}\left[\ln\frac{\sum_{y'}\exp(\langle\theta_t,\phi(x,y')\rangle)}{\sum_{y'}\exp(\langle\theta^\star,\phi(x,y')\rangle)}\left(\phi(x,y)-\mathbb{E}_{y'\sim\pi_{\theta_t}(\cdot|x)}[\phi(x,y')]\right)\right]$$

$$=\mathbb{E}_x\left[\ln\frac{\sum_{y'}\exp(\langle\theta_t,\phi(x,y')\rangle)}{\sum_{y'}\exp(\langle\theta^\star,\phi(x,y')\rangle)}\mathbb{E}_{y\sim\pi_{\theta_t}(\cdot|x)}\left(\phi(x,y)-\mathbb{E}_{y'\sim\pi_{\theta_t}(\cdot|x)}[\phi(x,y')]\right)\right]=0.$$

# G  Proof of Theorem 1

Our proof consists of three steps. First, we characterize the estimation error of $\widehat{V}^\star$. Second, with the learning guarantee of NR, we derive an upper bound on the average KL divergence between the output policies and the optimal policy $\pi^\star$ with DC. Finally, we apply Pinsker's inequality [Cover, 1999] to relate the KL divergence to the performance gap, completing the proof.

**Step 1.** To begin with, we can quantify the estimation error with Lemma 5:

**Lemma 5.** *With probability at least $1 - \delta/2$, we have for all $x$ that*

$$\left(\widehat{V}^\star(x) - V^\star(x)\right)^2 \lesssim \left(\beta \min\left\{\exp\left(\frac{1}{\beta}\right) - 1, \frac{1}{v_{\mathrm{ref}}}\right\}\right)^2 \frac{\ln(|\mathcal{X}|/\delta)}{N}.$$

The proof is deferred to Appendix G.1.

**Step 2.** We first bound $\sum_{t=1}^T \widehat{\ell}_t(\pi^\star)$ with Lemma 5. From the definition of $\pi^\star$, we know

$$\widehat{\ell}_t(\pi^\star) = \left(\widehat{V}^\star(x_i) - V^\star(x_i)\right)^2.$$

Let $\mathcal{F}_t$ denote the filtration induced by $\{x_1, y_1, \cdots, x_{t-1}, y_{t-1}\}$. Note that we have

$$\mathbb{E}\left[\widehat{\ell}_t(\pi^\star)\big|\mathcal{F}_t\right] = \mathbb{E}_{x\sim\rho}\left[\left(\widehat{V}^\star(x) - V^\star(x)\right)^2\right],$$

$$\mathbb{E}\left[\left(\widehat{\ell}_t(\pi^\star)\right)^2\Big|\mathcal{F}_t\right] = \mathbb{E}_{x\sim\rho}\left[\left(\widehat{V}^\star(x) - V^\star(x)\right)^4\right]$$

$$\leq \mathbb{E}_{x\sim\rho}\left[\left(\widehat{V}^\star(x) - V^\star(x)\right)^2\right] = \mathbb{E}\left[\widehat{\ell}_t(\pi^\star)\big|\mathcal{F}_t\right]. \tag{6}$$

Next we utilize the following form of Freedman's inequality [Song et al., 2022] from the literature:

**Lemma 6** (Freedman's inequality [Song et al., 2022])**.** *Let $\{Z_1, \ldots, Z_T\}$ be a sequence of non-negative random variables where $z_t$ is sampled from a distribution depending on $z_{1:t-1}$, i.e., $z_t \sim \mathbb{P}_t(z_{1:t-1})$. Further, suppose that $|Z_t| \leq R$ for all $t$. Then, for any $\delta > 0$ and $\lambda \in [0, \frac{1}{2R}]$, we have with probability at least $1 - \delta$ that,*

$$\left|\sum_{t=1}^T Z_t - \mathbb{E}[Z_t \mid \mathbb{P}_t]\right| \leq \lambda \sum_{t=1}^T \left(2R\mathbb{E}[Z_t \mid \mathbb{P}_t] + \mathbb{E}[Z_t^2 \mid \mathbb{P}_t]\right) + \frac{\ln(2/\delta)}{\lambda}.$$

Therefore, applying Freedman's inequality with Eq. (6), we have with probability at least $1 - \delta/4$ that

$$\sum_{t=1}^T \left(\widehat{V}^\star(x_i) - V^\star(x_i)\right)^2 \lesssim 2T\mathbb{E}_{x\sim\rho}\left[\left(\widehat{V}^\star(x) - V^\star(x)\right)^2\right] + \ln(1/\delta).$$

From Lemma 5, this implies we have with probability at least $1 - \delta/4$ that

$$\sum_{t=1}^T \widehat{\ell}_t(\pi^\star) \lesssim \left(\beta \min\left\{\exp\left(\frac{1}{\beta}\right) - 1, \frac{1}{v_{\mathrm{ref}}}\right\}\right)^2 \frac{T \ln(|\mathcal{X}|/\delta)}{N}. \tag{7}$$

On the other hand, from the definition of $\pi^\star$, we have

$$\mathbb{E}_{\pi\sim q_t}\left[\widehat{\ell}_t(\pi)\right] = \mathbb{E}_{\pi\sim q_t}\left[\left(\beta \ln \frac{\pi(y_t|x_t)}{\pi^\star(y_t|x_t)} + \left(\widehat{V}^\star(x_t) - V^\star(x_t)\right)\right)^2\right].$$

Note that we have

$$\mathbb{E}\left[\mathbb{E}_{\pi\sim q_t}\left[\widehat{\ell}_t(\pi)\right]\Big|\mathcal{F}_t\right] = \mathbb{E}_{x\sim\rho,\pi'\sim q_t, y\sim\pi'(\cdot|x)}\mathbb{E}_{\pi\sim q_t}\left[\left(\beta \ln \frac{\pi(y|x)}{\pi^\star(y|x)} + \left(\widehat{V}^\star(x) - V^\star(x)\right)\right)^2\right],$$

$$\mathbb{E}\left[\left(\mathbb{E}_{\pi\sim q_t}\left[\widehat{\ell}_t(\pi)\right]\right)^2\Big|\mathcal{F}_t\right] \leq \mathbb{E}_{x\sim\rho,\pi'\sim q_t, y\sim\pi'(\cdot|x)}\mathbb{E}_{\pi\sim q_t}\left[\left(\beta \ln \frac{\pi(y|x)}{\pi^\star(y|x)} + \left(\widehat{V}^\star(x) - V^\star(x)\right)\right)^4\right].$$

Note that from Assumption 2, we know

$$\left| \beta \ln \frac{\pi(y|x)}{\pi^\star(y|x)} + \left( \widehat{V}^\star(x) - V^\star(x) \right) \right| \le C_l + 1.$$

This implies that

$$\mathbb{E}\left[ \left( \mathbb{E}_{\pi \sim q_t} \left[ \widehat{\ell}_t(\pi) \right] \right)^2 \Big| \mathcal{F}_t \right] \le (C_l + 1)^2 \, \mathbb{E}\left[ \mathbb{E}_{\pi \sim q_t} \left[ \widehat{\ell}_t(\pi) \right] \Big| \mathcal{F}_t \right].$$

Therefore, with Freedman's inequality, we have with probability $1 - \delta/4$ that

$$\sum_{t=1}^{T} \mathbb{E}_{x \sim \rho, \pi' \sim q_t, y \sim \pi'(\cdot|x)} \mathbb{E}_{\pi \sim q_t} \left[ \left( \beta \ln \frac{\pi(y|x)}{\pi^\star(y|x)} + \left( \widehat{V}^\star(x) - V^\star(x) \right) \right)^2 \right]$$

$$\lesssim 2 \sum_{t=1}^{T} \mathbb{E}_{\pi \sim q_t} \left[ \widehat{\ell}_t(\pi) \right] + C_l^2 \ln(1/\delta). \tag{8}$$

Combining Eqs. (7) and (8) with the guarantee of no-regret oracle NR, we have with probability at least $1 - \delta$ that

$$\frac{1}{T} \sum_{t=1}^{T} \mathbb{E}_{x \sim \rho, \pi' \sim q_t, y \sim \pi'(\cdot|x)} \mathbb{E}_{\pi \sim q_t} \left[ \left( \beta \ln \frac{\pi(y|x)}{\pi^\star(y|x)} + \left( \widehat{V}^\star(x) - V^\star(x) \right) \right)^2 \right]$$

$$\lesssim \left( \beta \min \left\{ \exp\left( \frac{1}{\beta} \right) - 1, \frac{1}{v_{\mathsf{ref}}} \right\} \right)^2 \frac{\ln(|\mathcal{X}|/\delta)}{N} + \frac{\mathsf{Reg}(T)}{T} + \frac{C_l^2 \ln(1/\delta)}{T}.$$

Note that from AM-GM inequality, we have for all $\pi$ that

$$\mathbb{E}_{x \sim \rho, \pi' \sim q_t, y \sim \pi'(\cdot|x)} \left[ \left( \beta \ln \frac{\pi(y|x)}{\pi^\star(y|x)} + \left( \widehat{V}^\star(x) - V^\star(x) \right) \right)^2 \right]$$

$$= \beta^2 \mathbb{E}_{x \sim \rho, \pi' \sim q_t, y \sim \pi'(\cdot|x)} \left[ \left( \ln \frac{\pi(y|x)}{\pi^\star(y|x)} \right)^2 \right] + 2\beta \mathbb{E}_{x \sim \rho, \pi' \sim q_t, y \sim \pi'(\cdot|x)} \left[ \left( \ln \frac{\pi(y|x)}{\pi^\star(y|x)} \right) \left( \widehat{V}^\star(x) - V^\star(x) \right) \right]$$

$$+ \mathbb{E}_{x \sim \rho} \left[ \left( \widehat{V}^\star(x) - V^\star(x) \right)^2 \right]$$

$$\ge \frac{\beta^2}{2} \mathbb{E}_{x \sim \rho, \pi' \sim q_t, y \sim \pi'(\cdot|x)} \left[ \left( \ln \frac{\pi(y|x)}{\pi^\star(y|x)} \right)^2 \right] - \mathbb{E}_{x \sim \rho} \left[ \left( \widehat{V}^\star(x) - V^\star(x) \right)^2 \right].$$

Therefore we have with probability at least $1 - \delta$ that

$$\frac{1}{T} \sum_{t=1}^{T} \mathbb{E}_{x \sim \rho, \pi \sim q_t, \pi' \sim q_t, y \sim \pi'(\cdot|x)} \left[ \left( \beta \ln \frac{\pi(y|x)}{\pi^\star(y|x)} \right)^2 \right]$$

$$\lesssim \left( \beta \min \left\{ \exp\left( \frac{1}{\beta} \right) - 1, \frac{1}{v_{\mathsf{ref}}} \right\} \right)^2 \frac{\ln(|\mathcal{X}|/\delta)}{N} + \frac{\mathsf{Reg}(T)}{T} + \frac{C_l^2 \ln(1/\delta)}{T}. \tag{9}$$

From the definition of decoupling coefficient (Definition 1), we know that

$$\frac{1}{T} \sum_{t=1}^{T} \mathbb{E}_{\pi \sim q_t} \mathbb{E}_{x \sim \rho} \left[ \mathsf{KL}(\pi(\cdot|x), \pi^\star(\cdot|x)) \right] = \frac{1}{T} \sum_{t=1}^{T} \mathbb{E}_{x \sim \rho, \pi \sim q_t, y \sim \pi(\cdot|x)} \left[ \ln \pi(y|x) - \ln \pi^\star(y|x) \right]$$

$$\le \frac{1}{T} \sum_{t=1}^{T} \sqrt{ \mathsf{DC}(\Pi) \mathbb{E}_{x \sim \rho, \pi \sim q_t, \pi' \sim q_t, y \sim \pi'(\cdot|x)} \left[ \left( \ln \frac{\pi(y|x)}{\pi^\star(y|x)} \right)^2 \right] }.$$

Apply Cauchy-Schwartz inequality and we have

$$\frac{1}{T} \sum_{t=1}^{T} \mathbb{E}_{\pi \sim q_t} \mathbb{E}_{x \sim \rho} \left[ \mathsf{KL}(\pi(\cdot|x), \pi^\star(\cdot|x)) \right] \le \sqrt{ \frac{\mathsf{DC}(\Pi)}{T} \sum_{t=1}^{T} \mathbb{E}_{x \sim \rho, \pi \sim q_t, \pi' \sim q_t, y \sim \pi'(\cdot|x)} \left[ \left( \ln \frac{\pi(y|x)}{\pi^\star(y|x)} \right)^2 \right] }. \tag{10}$$

Combine Eqs. (9) and (10) and we have with probability at least $1 - \delta$ that

$$
\frac{1}{T} \sum_{t=1}^{T} \mathbb{E}_{\pi \sim q_t} \mathbb{E}_{x \sim \rho} \left[ \mathsf{KL}(\pi(\cdot|x), \pi^\star(\cdot|x)) \right]
$$

$$
\lesssim \left( \min \left\{ \exp \left( \frac{1}{\beta} \right) - 1, \frac{1}{v_{\mathsf{ref}}} \right\} \right) \sqrt{\frac{\mathsf{DC}(\Pi) \ln(|\mathcal{X}|/\delta)}{N}} + \sqrt{\frac{\mathsf{DC}(\Pi)\mathsf{Reg}(T)}{T\beta^2}} + \sqrt{\frac{\mathsf{DC}(\Pi)C_l^2 \ln(1/\delta)}{T\beta^2}}.
$$
(11)

**Step 3.** Note that from Pinsker's inequality [Cover, 1999], we have

$$
J(\pi^\star) - \frac{1}{T} \sum_{t=1}^{T} \mathbb{E}_{\pi \sim q_t}[J(\pi)] = \mathbb{E}_{x \sim \rho, y \sim \pi^\star(\cdot|x)}[r(x,y)] - \frac{1}{T} \sum_{t=1}^{T} \mathbb{E}_{\pi \sim q_t, x \sim \rho, y \sim \pi(\cdot|x)}[r(x,y)]
$$

$$
\leq \frac{1}{T} \sum_{t=1}^{T} \mathbb{E}_{\pi \sim q_t} \mathbb{E}_{x \sim \rho} \left[ \| \pi(\cdot|x) - \pi^\star(\cdot|x) \|_1 \right] \leq \frac{1}{T} \sum_{t=1}^{T} \mathbb{E}_{\pi \sim q_t} \mathbb{E}_{x \sim \rho} \left[ \sqrt{2\mathsf{KL}(\pi_t(x), \pi^\star(\cdot|x))} \right].
$$

Therefore from Cauchy-Schwartz inequality, we have

$$
J(\pi^\star) - \frac{1}{T} \sum_{t=1}^{T} \mathbb{E}_{\pi \sim q_t}[J(\pi)] \leq \sqrt{\frac{1}{T} \sum_{t=1}^{T} \mathbb{E}_{x \sim \rho} \left[ 2\mathsf{KL}(\pi_t(x), \pi^\star(\cdot|x)) \right]}.
$$
(12)

Combine Eqs. (11) and (12) and we have with probability at least $1 - \delta$ that

$$
J(\pi^\star) - \frac{1}{T} \sum_{t=1}^{T} \mathbb{E}_{\pi \sim q_t}[J(\pi)] \lesssim \left( \frac{\mathsf{DC}(\Pi)\mathsf{Reg}(T)}{T\beta^2} \right)^{\frac{1}{4}} + \left( \frac{\mathsf{DC}(\Pi)C_l^2 \ln(1/\delta)}{T\beta^2} \right)^{\frac{1}{4}}
$$

$$
+ \left( \left( \min \left\{ \exp \left( \frac{1}{\beta} \right) - 1, \frac{1}{v_{\mathsf{ref}}} \right\} \right)^2 \frac{\mathsf{DC}(\Pi) \ln(|\mathcal{X}|/\delta)}{N} \right)^{\frac{1}{4}}.
$$

This concludes our proof.

### G.1 Proof of Lemma 5

Let $\widehat{p}(x)$ denote the empirical average reward that $\pi_{\mathsf{ref}}$ attains for prompt $x$ in the offline estimation stage. Then we can observe that

$$
\widehat{V}^\star(x) = \beta \ln \left( (1 - \widehat{p}(x)) + \widehat{p}(x) \exp(1/\beta) \right).
$$

We also use $p^\star(x)$ to denote $\mathbb{E}_{y \sim \pi_{\mathsf{ref}}(x)}[r(x,y)]$. From Azuma-Hoeffding's inequality [Azuma, 1967] and union bound over all $x \in \mathcal{X}$, with probability at least $1 - \delta/2$, we have for all $x \in \mathcal{X}$ that

$$
|\widehat{p}(x) - p^\star(x)| \lesssim \sqrt{\frac{\ln(|\mathcal{X}|/\delta)}{N}}.
$$

On the other hand, we know

$$
V^\star(x) = \beta \ln \left( (1 - p^\star(x)) + p^\star(x) \exp(1/\beta) \right), \quad \forall x.
$$

From mean value theorem, we have

$$
\left| \widehat{V}^\star(x) - V^\star(x) \right| \leq \beta \left( \min \left\{ \exp \left( \frac{1}{\beta} \right) - 1, \frac{1}{v_{\mathsf{ref}}} \right\} \right) |\widehat{p}(x) - p^\star(x)|, \quad \forall x.
$$

Therefore, we have for all $x \in \mathcal{X}$

$$
\left( \widehat{V}^\star(x) - V^\star(x) \right)^2 \lesssim \left( \beta \min \left\{ \exp \left( \frac{1}{\beta} \right) - 1, \frac{1}{v_{\mathsf{ref}}} \right\} \right)^2 \frac{\ln(|\mathcal{X}|/\delta)}{N}.
$$

# H Additional Related Works

**RL for LLM Reasoning** Reinforcement learning (RL) has become a standard approach for post-training large language models (LLMs). The most prominent example is Reinforcement Learning from Human Feedback (RLHF), popularized by Christiano et al. [2017] and Touvron et al. [2023], which has garnered significant attention from both industry and academia due to its remarkable success in aligning LLM behavior with human preferences. While this two-stage framework has achieved impressive results, it remains computationally inefficient, requiring separate training of both a reward model and a policy model. To mitigate this overhead, recent work has focused on developing one-stage algorithms, including SLiC [Zhao et al., 2023], DPO [Rafailov et al., 2024], REBEL [Gao et al., 2024a], and its variants such as IPO [Azar et al., 2024], SPO [Swamy et al., 2024], SPPO [Wu et al., 2024], and DRO [Richemond et al., 2024]. Building on recent advances from OpenAI O1 [OpenAI, 2024] and DeepSeek R1 [DeepSeek-AI, 2025], Reinforcement Learning with Verifiable Rewards (RLVR) has gained traction as an effective approach for improving LLM reasoning capabilities, particularly in domains such as mathematics and programming. Notable examples include GRPO [Shao et al., 2024], DR-GRPO [Liu et al., 2025a], DAPO [Liu et al., 2024], OREO [Wang et al., 2024], DQO [Ji et al., 2024], and RAFT++ [Xiong et al., 2025]. Similar to OREO, DAPO, DQO, and DRO, $A^\star$-PO operates efficiently by requiring only a single response per prompt and regressing the log ratio of $\pi$ and $\pi_{\mathrm{ref}}$ to the advantage. However, unlike these methods, $A^\star$-PO eliminates the need to train an explicit critic for value estimation, further improving its computational efficiency.

**Exploration in RL** Exploration is a critical component of reinforcement learning, particularly in complex reasoning tasks where large language models must navigate vast response spaces to discover correct solutions and learn from them [Dwaracherla et al., 2024]. In classic online RL methods such as PPO [Schulman et al., 2017] and GRPO [Shao et al., 2024], exploration is typically encouraged through entropy regularization. By maximizing policy entropy, these methods discourage overly deterministic behavior, promoting more diverse outputs. However, this heuristic approach often leads to suboptimal policies and slow convergence due to increased objective complexity. To address these limitations, recent works have explored active exploration strategies [Xiong et al., 2024, Ye et al., 2024, Xie et al., 2024, Cen et al., 2024, Zhang et al., 2024, Bai et al., 2025, Chen et al., 2025a], which estimate environment uncertainty from historical data and plan optimistically. While these methods provide theoretical guarantees on sample efficiency, they often rely on complex exploration mechanisms or strong assumptions—such as reward linearity—making them difficult to implement and resulting in unstable empirical performance.

In contrast, $A^\star$-PO does not perform explicit exploration, resulting a much simpler algorithm. $A^\star$-PO is designed for a more realistic setting, where we do not expect RL post-training can help solve problems of which pass@K values are zero for some large K under $\pi_{\mathrm{ref}}$. Recent studies [Zhao et al., 2025, Yue et al., 2025] suggest that existing RL algorithms struggle to incentivize capabilities that go beyond the knowledge boundaries of the pretrained model—failing, for instance, to surpass the Pass@K performance achievable by the base model $\pi_{\mathrm{ref}}$. As shown in Section 5, we prove that as long as we do not wish RL post-training to solve problems where $\pi_{\mathrm{ref}}$ has **zero** probability of solving, even without any explicit exploration mechanism, $A^\star$-PO provably converges to $\pi^\star$ with polynomial sample complexity. In this regard, $A^\star$-PO achieves both training efficiency and optimality under a realistic setting.

Prior work Q# [Zhou et al., 2025] also leverages such a condition to achieve a near optimal regret bound for the KL-regularized RL objective at token level without performing any exploration. The difference is that Q# operates at the token level and learns a critic Q function to guide the base model to perform controllable generation at test time. $A^\star$-PO is a direct policy optimization approach. Another subtle difference in theory is that by operating at the trajectory-level, $A^\star$-PO's sample complexity does not have a polynomial dependence on the size of the vocabulary. This is because when we operate at the trajectory-level, we can query the ground truth reward model to get a perfect reward for a given trajectory (e.g., whether the response is correct). The idea of directly regressing optimal advantage $A^\star$ is motivated from the classic imitation learning algorithm AggreVaTe(d) [Ross and Bagnell, 2014, Sun et al., 2017] where these algorithm aim to learn a policy that greedily selects actions by maximizing the expert policy's advantage function. The difference is that $A^\star$-PO operates in the pure RL setting where one does not have an expert to query for feedback.

# I  Rate of Convergence

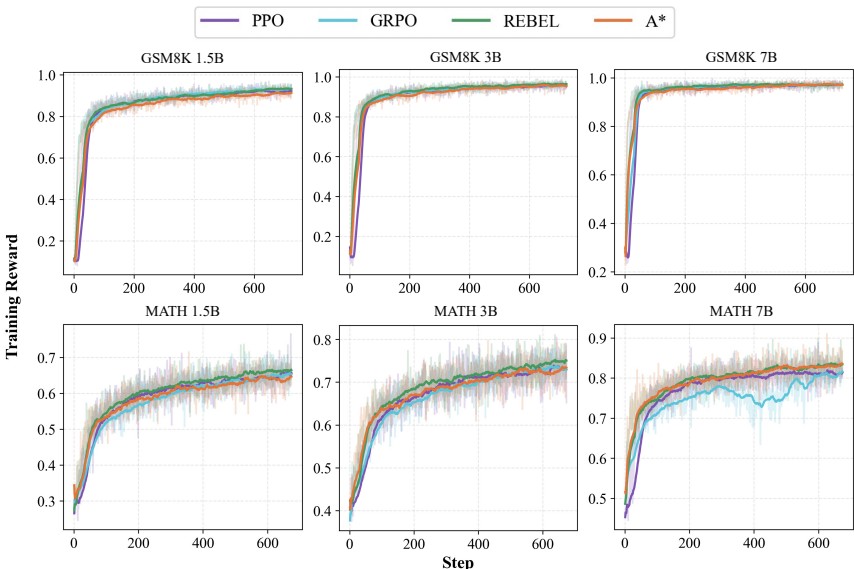

Figure 11: **Training reward vs. steps over two datasets and three models.**

Fig. 11 shows the training reward of PPO, GRPO, REBEL, and $A^\star$-PO over MATH and GSM8K datasets at each step. We can see that all four methods have similar convergence behavior. Based on the plots, we can now safely conclude that $A^\star$-PO is the fastest method as it requires the least amount of time to complete the same number of training steps, while reaching convergence at the same step count as the other methods. To illustrate this more clearly, we also include a plot of training and generation time versus training reward below. Note that the $A^\star$-PO curves do not start at time zero, as they include the time required for data generation. Each line starts at the corresponding generation time for its dataset and model. While REBEL converges at a comparable rate in terms of time, it requires significantly more memory than $A^\star$-PO.

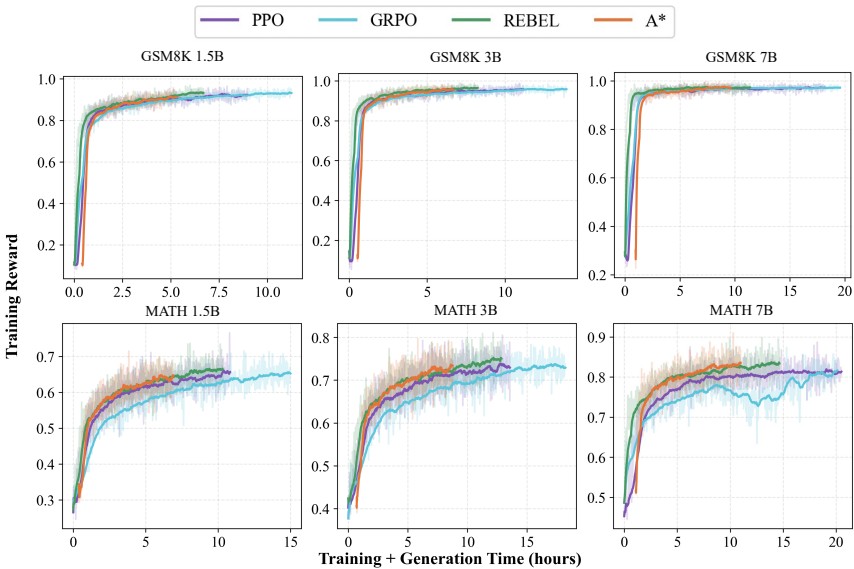

Figure 12: **Training reward vs. training and generation time over two datasets and three models.**

