# OpenReview forum: "Accelerating RL for LLM Reasoning with Optimal Advantage Regression"
_NeurIPS.cc/2025/Conference — NeurIPS 2025 poster_

### Official Review · Reviewer_q8X3 · 2025-07-01

**Clarity:** 2
**Significance:** 2
**Originality:** 2
**Rating:** 4
**Confidence:** 5

**Summary:**

The paper introduces A⋆-PO (Policy Optimisation via Optimal Advantage Regression), a two-stage RL procedure for post-training LLMs on reasoning tasks. The work offers a neat regression-based alternative to critic-heavy RL algorithms, with solid speed-ups and "formal guarantees", but those guarantees require many unrealistic assumptions that may not hold for frontier-scale LLMs.

I would like to clarify a point before I move on to the questions. it is known that under sufficient assumptions, one can get vanishing regrets; however the active area of research is generalising those to problems that include non-linearities and non-convexity.

**Questions:**

Please see above

**Ethical Concerns:**

["NO or VERY MINOR ethics concerns only"]

**Final Justification:**

Rebuttal addressed my concerns!

**Limitations:**

Briefly mentioned

**Quality:**

2

**Strengths And Weaknesses:**

The paper has a nice flow to it and reads well, till the theoretical section which I felt was not very well presented. I feel for the authors due to the constraints of space, but more discussions on the assumptions and their impact are indeed needed. I would like to ask the authors some questions:

1. When are policy classes not misspecified? Because if they are, then the realizability assumption would have issues. Please discuss the approximation error or clearly detail which architectures make the assumption plausible and why?

2. In assumption 2, can the log-ratio explode? I thought it could, e.g, if we pair a small model with a larger reference one? Is this correct? Can the authors please help me understand? If so, I think the theory should handle how to renormalise?

3. In assumption 3, v_ref can be very low, especially on some hard problems. This might make the bound vacuous, can the authors please characterise performance as v_ref tends to zero? Maybe an adaptive sampling procedure might help to remedy this issue?

4. The Fisher assumption also doesn't really hold in many high-dimensional cases with deep networks - in fact, many times the Fisher matrix ends up ill-conditioned. Maybe the authors could add a regularisation (which will change the update rules) and study and quantify the value of such regularisers needed in practice.

5. The Gap hides a massive constant. Can the authors please bound those constants or give ballpark numbers so that those aren't massive, like they are not > 1?

I don't mean to be annoying, but given that I come from a theoretical background, the value of the theoretical section in this paper is minimal from my point of view; those assumptions are standard and (almost always and this is a well known fact) do not apply to non-convex losses, deep networks, and transformers. In other words, it is like taking a theory that had been developed for some types of problems and adding it to a new type of problem, but making the relevant assumptions so that the new type of problem meets the first type.

6. I would like to see more experiments related to coding benchmarks as well; they also require formal reasoning. Can the authors please report those if possible?

7. Please publish wall-clock times per stage and GPU hours for all baselines.

8. There are no error bars, and the authors clearly say NO to the statistical significance question. I am not sure how to take that? Like are the results reported in the paper by luck?

I am happy to improve my scores of the above questions have been answered.

---

> ### Author Rebuttal · Authors · 2025-07-29
>
> We thank the reviewer for their detailed comments.
>
> > the active area of research is generalising those to problems that include non-linearities and non-convexity.
>
> We acknowledge this point, and would like to clarify that our theoretical analysis is exactly on **nonconvex and nonlinear cases**. More specifically, **Theorem 1 addresses general policy classes**, which are nonconvex and nonlinear. Consequently, the loss function $\widehat{l}_t$ is also **nonconvex and nonlinear with respect to $\pi$**. To handle this, we introduce the decoupling coefficient (Definition 1), which is effective when dealing with nonconvexity and nonlinearity.
>
> We also present a case study on log-linear policy classes to illustrate our theoretical results more clearly. We would like to clarify that **log-linear policy classes and the corresponding loss functions are still “nonconvex and nonlinear” with respect to the parameter $\theta$**. We designed two specific algorithms (utilizing FTPL and OGD) for this case and they achieve sub-linear regret. We will further highlight our focus on nonconvexity in the paper.
>
> > Weaknesses 1
>
> We assume policy class realizability to simplify presentation, as almost all the existing theoretical works do. That said, **our analysis can be generalized to misspecified policy classes easily**. Assume that the approximation error of the policy class is as follows:
> $$\min_{\pi\in\Pi, x, y} |\ln\pi(y|x)-\ln\pi^{\star}(y|x)|\leq C_{\mathsf{approx}}.$$
> Let $\widetilde{\pi}$ denote the policy in $\Pi$ that is closest to $\pi^{\star}$. Then we have
> $$\widehat{l} ^t(\widetilde{\pi}) \leq 2\widehat{l} ^t(\pi^{\star}) + 2\beta^2C^2_{\mathsf{approx}}.$$
> Substituting the above inequality to the proof in Appendix H, we know that working with misspecified policy classes is equivalent to adding $\beta^2C^2_{\mathsf{approx}}$ to the regret. All the results still hold, with just an additional approximation error.
>
> > Weaknesses 2
>
> We would like to point out that in empirical works, with clipping techniques, such log ratios can be controlled and not blow up. That is also the reason why the existing theoretical works on DPO-type algorithms have similar assumptions.
>
> > Weaknesses 3
>
> First we would like to clarify that our bound depends on $min(\exp(1/\beta), v_{ref})$. Therefore, if $\beta$ is fixed (which is typically the setting in practice), our result would not be worsened by small $v_{ref}$. On the other hand, a well known and widely applied technique in LLM training is filtration, i.e., filtering out the prompts that are too difficult or too easy. With this process, $v_{ref}$ will be reasonably large during RL training, preventing $1/v_{ref}$ from blowing up.
>
> > Weaknesses 4
>
> First, Fisher matrix assumption is not necessary in terms of achieving sublinear regret. Corollary 1 does not need any assumptions on Fisher matrix. The reason why we introduce Fisher matrix assumption is that we want to claim that our algorithm can enjoy a faster convergence when the problem is regular and well conditioned. That said, in our experiments, we used OGD and our algorithm works pretty well. This demonstrates that even if the Fisher matrix can be ill conditioned sometimes, the problem is often regular and OGD can work in practice.
>
> > Weaknesses 5
>
> Through a rough calculation, the constants in Theorems 1 and 2 can essentially be bounded by 16 and 64, respectively. For Theorem 1, we omitted four constants of 2 in Lines 680, 681, 686, and 690. For Theorem 3, the constant on the right-hand side of the inequality in Line 651 can be bounded by 256, which implies that the constant in Line 655 is $4\sqrt{256} = 64$. Notice that the actual constants will be much smaller than these theoretical values, especially considering that we relaxed these constant terms for analytical convenience in analysis. Even so, it suffices to say that these constants are not massive.
>
> > those assumptions are standard and do not apply to non-convex losses, deep networks, and transformers
>
> We agree that these assumptions are standard, but we respectively disagree that they do not apply to these settings. First, **realizability does not depend on whether the function classes are convex. Instead, more complicated function classes like deep networks and transformers typically have a higher representation capacity and thus realizability is easier to satisfy (at least the approximation error would be smaller.)** Second, **other assumptions have their origins in practice.** The log ratio is bounded because of clipping, and $v_{ref}$ is not small because of filtration. In addition, these techniques work well in practice, which further support the validity of these assumptions. **Since these assumptions are standard in existing works, we do not assume unreasonable assumptions to simplify analysis or use other works’ results directly.**
>
> > it is like taking a theory that had been developed for some types of problems and adding it to a new type of problem, but making the relevant assumptions so that the new type of problem meets the first type
>
> We respectfully disagree with this claim. As the reviewer has noted, our assumptions are standard in existing works, indicating that we do not introduce additional, unreasonable assumptions merely to reuse existing theory. On the contrary, **our analysis introduces several novel techniques and results.** We summarize them below for clarity:
> - **Getting rid of policy completeness assumption.** In our analysis we only need policy realizability. In contrast, other on-policy policy optimization approaches including PG, PPO and REBEL rely on a stronger condition called policy completeness. We avoid such a condition **by designing novel loss functions, which directly regress to the optimal policy rather than next-step policies similar to policy iteration.**
> - **Getting rid of exploration mechanisms.** Our method does not incorporate any explicit exploration mechanism such as optimism in the face of uncertainty or reset to some exploratory distribution as in existing works. **This is a significant improvement because exploration is necessary in general RL settings.** We achieve this by noticing that in the context of LLMs, **there exists a close connection between the novel loss functions we proposed and the KL divergence between the learned policy and the optimal policy** (illustrated in step 2 of the proof in Appendix H). **This is a new finding.** Leveraging this observation, we managed to bound the suboptimality of the learned policy efficiently.
> - **New findings on nonconvex no-regret  learning with OGD.** In the case study of log-linear policy classes, we find that OGD provably converges with the optimal rate. **This is a new finding because the loss functions are still nonconvex with log-linear policy classes and OGD does not work for nonconvex losses in general.** We achieve such a result not by making unreasonable assumptions, but by **establishing a connection between the gradient of the loss and the distance from the optimal parameter** (Lemma 3 in Appendix G), **which is a new finding in this area.** This connection enables us to efficiently bound the distance between the learned policy and optimal policy. Notably, the techniques used in Appendix G are totally different from the ones we use in Appendix H (Theorem 1), which is the reason why we can obtain a faster convergence rate.
>
> > Weaknesses 6
>
> We agree that coding benchmarks are an important and challenging class of reasoning tasks. However, we did not include them in this work due to the significant computational overhead involved—specifically, reward computation typically requires running unit tests in isolated environments, which introduces substantial cost and engineering complexity. Despite this, we believe $A^\star$-PO remains well-suited for coding tasks, as it only requires one sample per prompt during online training, and thus demands fewer reward queries than approaches like GRPO that rely on multiple generations. This efficiency advantage would likely persist in the coding domain, and we are excited to explore this direction in future work.
>
> > Weaknesses 7
>
> Thank you for the suggestion. We include wall-clock training times for each method in Figure 2 and Tables 1 and 2. As described in Appendix A.2 and A.4, our experiments were run on machines with 4 H100 GPUs (for Sections 4.1 and 4.2) and 8 H100 GPUs (for Section 4.3). Therefore, GPU hours can be directly computed by multiplying the reported training times by the number of GPUs used. To further clarify the training efficiency of $A^\star$-PO, we include a rate-of-convergence analysis in the supplemental material, showing training and generation time versus reward. We will move this analysis into the appendix in the camera-ready version to make this information more accessible.
>
> > Weaknesses 8
>
> We appreciate the opportunity to clarify. While we did not include error bars in the current submission, all experiments were run with fixed seeds, making our results exactly reproducible. Furthermore, in large-scale RL for LLMs, empirical results tend to be stable across runs due to the high sample count per iteration—often involving hundreds of prompt-response pairs per update and training over thousands of steps. To further reduce variance, we report results on large-scale datasets and, for smaller benchmarks such as AMC 23, AIME 24/25, and HMMT Feb 24/25, we compute the average performance over 32 generations per prompt. This setup ensures statistical reliability even in the absence of formal error bars. We are confident that the observed performance improvements are robust and not due to chance, as they are consistent across models, datasets, and evaluation metrics.

---

> > ### Comment · Reviewer_q8X3 · 2025-08-04
> > **Thank you**
> >
> > Thank you for the clarification. It makes sense.

---

> > > ### Author Response · Authors · 2025-08-05
> > >
> > > Thank you for the timely response! Do you have any further questions or points you would like us to clarify? We would be more than happy to help.

---

### Official Review · Reviewer_bvzh · 2025-07-02

**Clarity:** 4
**Significance:** 3
**Originality:** 3
**Rating:** 5
**Confidence:** 4

**Summary:**

This work proposes $A^*$-PO, an RL technique to finetune reasoning capabilities with improved sample efficiency. It achieves this by first learning an optimal advantage function offline on generations produced by the base reference policy, then during online training, regressing the log ratio against it. The authors demonstrate the improved efficiency compared to PPO/GRPO with only one generation per prompt during online training.

**Questions:**

These are some specific questions I had in addition to the ones I listed above:
- In Table 1, what is the significance of showing lower KL-divergence from the base policy? Also, would this not be also impacted by the choice of $\beta$ in baseline methods?
- Have the authors considered adaptively changing $N$ based on problem difficulty? For simpler problems, a larger $N$ might not be necessary to estimate the advantage function, whereas for harder problems, it could be more crucial.
- How well does a model post-trained with $A^{*}$-PO retain its original abilities, i.e., does it exhibit more/less catastrophic forgetting?

**Ethical Concerns:**

["NO or VERY MINOR ethics concerns only"]

**Final Justification:**

Here is a summary of my thoughts after the rebuttal:
* The generalization of the method to arbitrary/dense rewards is clear and promising.
* Filtering strategy: Given that filtering is enabled by offline sampling in the proposed method, I asked a tiny follow-up question: does this filtering lead to any bias in the 'online' training distribution, especially in domains where initial success rates are low?
* I encouraged the authors to discuss any interactions between $N$ and $\beta_1$, given that both impact the quality of value estimates.
* The empirical and theoretical results in App D/E make sense.

Overall, the responses address my initial questions well, and I would like to maintain my positive rating and increase my confidence.

**Limitations:**

Yes.

**Paper Formatting Concerns:**

NA.

**Quality:**

3

**Strengths And Weaknesses:**

**Strength**
- The proposed method is novel and theoretically grounded. The paper theoretically and empirically supports the improved sample complexity.
- The experiments are well-designed and comprehensive, including evaluations on GSM8K, MATH, and long-context benchmarks, as well as showing generalization to other benchmarks after training.

**Weaknesses**
I am not an RL expert, and would defer constructive criticism to more experienced reviewers. I mainly have a few questions:
- How would the estimation of $V^*$ perform with non-binary or denser reward signals (e.g., with process-supervision)?
- The authors state that during online training, only one sample is required to regress on the estimated optimal advantage, what would be the impact if multiple samples were generated?
- Clarifying question: In Fig 4, is filtering applied during the online stage, i.e., problems that did not find a correct solution (as measured in Stage 1) are not used for online training? Would this technique also apply to PPO/GRPO, or is this something specific to your method?
- Have the authors ablated the effect of different values of $\beta$ for the two stages?

---

> ### Author Rebuttal · Authors · 2025-07-29
>
> We thank the reviewer for their constructive and thoughtful comments.
>
> > How would the estimation of $V^\star$ perform with non-binary or denser reward signals (e.g., with process-supervision)?
>
> Our method is compatible with non-binary and dense rewards, as we make no assumptions about the reward values. While our experiments focus on the bandit setting with terminal rewards, the value estimation naturally extends to dense or process-level supervision. Consider a setting with a prompt $x$ and a response with $H$ steps $y_{1:H}$. Then the optimal value at step $h$ can be expressed as $V^\star(x, y_{1:h}) = \beta \ln \mathbb{E} {\pi_{ref}} \left[\exp( \sum_{t > h} r(x,y_{1:t}) / \beta )\right]$ where $r(x,y_{1:t})$ is the process reward for step $t$. Thus, our framework remains applicable under denser reward feedback. We leave the empirical exploration under these settings for future work.
>
> > The authors state that during online training, only one sample is required to regress on the estimated optimal advantage, what would be the impact if multiple samples were generated?
>
> Theoretically, using multiple samples per prompt has no effect on the expected loss as $ \mathbb{E}{x \sim \rho, y \sim \pi(\cdot|x)}[\ell(x, y)] = \mathbb{E}{x \sim \rho,(y_1, \dots, y_K) \sim \pi(\cdot|x)}\left[\frac{1}{K} \sum_{k=1}^K \ell(x, y_k)\right] $ holds for any loss function $\ell$ and any $K$. However, in practice, generating multiple samples per prompt—as in GRPO—increases training time and memory usage. Our method is more efficient because it achieves comparable performance with only a single generation per prompt.
>
> > Clarifying question: In Fig 4, is filtering applied during the online stage, i.e., problems that did not find a correct solution (as measured in Stage 1) are not used for online training? Would this technique also apply to PPO/GRPO, or is this something specific to your method?
>
> Yes, the filtering is applied before the online stage: we discard prompts for which none of the $N$ offline samples from $\pi_{\text{ref}}$ yield a correct answer. While this strategy could be applied to PPO or GRPO, those methods do not have offline sampling and would need to perform additional inference solely for filtering, which adds further overhead to already more complex training pipelines. In contrast, our method naturally performs offline sampling for value estimation, so filtering introduces negligible extra cost and offers a natural efficiency bonus.
>
> > Have the authors ablated the effect of different values of $\beta$ for the two stages?
>
> Yes, this is addressed in Appendix D. As $\beta_1 \to \infty$, the estimated $V^\star(x)$ approaches the expected reward under $\pi_{\text{ref}}$; as $\beta_1 \to 0$, it converges to the $\text{Pass@}N$ performance. These properties are theoretically proven in Appendix E.
>
> We choose a larger $\beta_1$ for the offline stage to provide a smooth and stable value estimate, and a smaller $\beta_2$ during online training to relax the KL constraint and promote reward maximization. Our ablations show that overly optimistic or pessimistic estimates of $V^\star(x)$ can both harm performance. For $\beta_2$, we found that $10^{-3}$ consistently gives strong results; larger or smaller values would degrade the performance.
>
> > In Table 1, what is the significance of showing lower KL-divergence from the base policy? Also, would this not be also impacted by the choice of $\beta$ in baseline methods?
>
> Lower KL divergence indicates that the trained policy stays closer to the reference policy while still achieving strong performance. This is significant because the KL-regularized RL objective aims to optimize reward while staying minimally divergent from the base model. Our results show that $A^\star$-PO induces significantly less change to the base model than PPO/GRPO for comparable or better performance—suggesting better stability and less catastrophic forgetting.
>
> Indeed, the KL divergence can be influenced by the choice of $\beta$ in baseline methods. However, we tuned $\beta$ for all methods to maximize the validation accuracy to ensure a fair comparison, and $A^\star$-PO consistently achieves the smallest KL divergence across model sizes and datasets.
>
> > Have the authors considered adaptively changing $N$ based on problem difficulty? For simpler problems, a larger $N$ might not be necessary to estimate the advantage function, whereas for harder problems, it could be more crucial.
>
> This is an excellent suggestion. Intuitively, prompts with intermediate success rates (e.g., around 50% accuracy under $\pi_{\text{ref}}$) have higher variance in their value estimates, and would benefit from more samples to improve estimation accuracy. In contrast, very easy or very hard prompts (i.e., near 0% or 100% success) require fewer samples, as their value estimates are more stable. While we have not explored adaptive selection of $N$ in this work, it is indeed a promising direction.
>
> > How well does a model post-trained with $A^\star$-PO retain its original abilities, i.e., does it exhibit more/less catastrophic forgetting?
>
> While we did not explicitly test for catastrophic forgetting, the significantly lower KL divergence to the base model suggests that $A^\star$-PO better preserves the original capabilities compared to PPO and GRPO. This is an important benefit for applications requiring general-purpose LLMs with minimal task-specific overfitting.

---

> > ### Comment · Reviewer_bvzh · 2025-08-05
> > **Thank you**
> >
> > I thank the authors for their detailed clarifications.
> >
> > * The generalization of the method to arbitrary/dense rewards is clear and promising.
> > * Filtering strategy: Given that filtering is enabled by offline sampling in your method, I have a tiny follow-up question: does this filtering lead to any bias in the 'online' training distribution, especially in domains where initial success rates are low?
> > * I would also encourage the authors to discuss any interactions between $N$ and $\beta_1$, given that both impact the quality of value estimates.
> > * The empirical and theoretical results in App D/E make sense.
> >
> > Overall, the responses address my initial questions well, and I would like to maintain my positive rating and increase my confidence. I look forward to seeing further work in these directions.

---

> > > ### Author Response · Authors · 2025-08-05
> > >
> > > Thank you for taking the time to read our rebuttal and maintaining the rating.
> > >
> > > > does this filtering lead to any bias in the 'online' training distribution, especially in domains where initial success rates are low?
> > >
> > > Recently, it has been shown that RLVR-trained models outperform their base models on pass@k performance at smaller values of k (e.g., k=1), while base models achieve a higher pass@k score when k is large [1] in math/coding/visual reasoning benchmarks. It suggests that RL reasoning abilities originate from and are bounded by the base model. In other words, problems that the reference policy is unable to solve are unlikely to be solved through RL post-training. Motivated by this insight and supported by our empirical results in Section 4.4, we believe that filtering does not introduce a significant bias or degrade online training performance in the domains we study. On the other hand, we agree that in settings where initial success rates are low and RL post-training leads to substantial performance gains, this assumption may no longer hold. In such cases, filtering could potentially limit the effectiveness of training. However, since our work focuses on LLM reasoning tasks, we believe filtering remains a safe and effective strategy in our context.
> > >
> > > > I would also encourage the authors to discuss any interactions between $N$ and $\beta_1$, given that both impact the quality of value estimates.
> > >
> > > Thank you for your feedback! We will make sure to discuss the effect of $N$ and $\beta_1$ in the main body of the revised version.
> > >
> > > [1] Yue, Y., Chen, Z., Lu, R., Zhao, A., Wang, Z., Song, S., & Huang, G. (2025). Does reinforcement learning really incentivize reasoning capacity in llms beyond the base model?. arXiv preprint arXiv:2504.13837.

---

### Official Review · Reviewer_yXFw · 2025-07-03

**Clarity:** 3
**Significance:** 3
**Originality:** 3
**Rating:** 5
**Confidence:** 4

**Summary:**

This work focuses on the topic of using reinforcement learning to enhance the LLMs' reasoning capability.  Previous RL methods either rely on training a critic network (e.g., PPO) or generating multiple samples per prompt (e.g., GRPO) during online finetuning. This work, based on the property of the optimal value function under the KL-regularized policy, proposes a two-stage method, where the first stage offline sampling is performed directly via the reference policy to estimate the optimal value function, and the second stage performs online training with on-policy updates using a simple least-square loss and one response per prompt. The effectiveness of the proposed method, i.e., A*PO is verified with experimental results, showing that it achieves comparable results with previous methods, while speeding up training and reducing memory usage. Furthermore, theoretical results are also provided, providing the theoretical soundness of a more general version of A*PO, without strong assumptions or using complex exploration strategies but achieving nice convergence results and yielding attractive properties.

**Questions:**

There are a few clarification questions that I have in mind to discuss with the authors:

- At the beginning of Section 4, there is a discussion on using two different KL-regularization coefficient during two stages. I am wishing to understand more about the choices and the reason behind. In particular, if I understand it correctly, using different beta essentially lead to different optimal values, which then damages the usefulness of the offline value estimation for the online phase.

- Regarding the theoretical innovation concern raised above, I am also looking forward to learning from the authors more about the underlying and probably hidden contributions there.

**Ethical Concerns:**

["NO or VERY MINOR ethics concerns only"]

**Final Justification:**

The authors have answered my questions and I would like to maintain my favorable consideration of this work due to its strong theoretical insights and empirical performances.

**Limitations:**

Yes

**Quality:**

3

**Strengths And Weaknesses:**

This work is a nice exploration of the more efficient methods in RL finetuning of LLM:

- The proposed method is based on theoretical properties of the optimal value under the KL-regularized target, in particular, discovering that it can be estimated via sampling through the reference policy

- The designed algorithm is clear, intuitive, and easy to implement. It tackles the problem that existing methods use extra resources (both computation and memory) to estimate the value function. The pre-computed value estimation from the offline sampling can save resource and time in online fine-tuning.

- The experimental results are comprehensive in terms of metrics. It helps a lot to visualize the comparison of time and memory used by different methods.

- The theoretical results are also solid. I really appreciate the nice connection between empirical designs and theoretical understandings. The results themselves can be valuable in understanding the KL-regularized RL/contextual bandits problem.

- The overall presentation is clear and well-organized.

If I have to nit-picking, I would hope the authors provide more illustrations of the novelty or innovation behind the theoretical discussions. Currently, the results are well-illustrated, but the underlying proof technics are largely deferred to the appendix without many highlights on the challenges/novelties there.

---

> ### Author Rebuttal · Authors · 2025-07-29
>
> We sincerely thank the reviewer for the thoughtful and detailed feedback.
>
> > If I have to nit-picking, I would hope the authors provide more illustrations of the novelty or innovation behind the theoretical discussions. Currently, the results are well-illustrated, but the underlying proof technics are largely deferred to the appendix without many highlights on the challenges/novelties there.
>
> Thank you for the suggestion. We will highlight the key theoretical innovations more explicitly in the main text in the camera-ready version. With the additional page for camera ready, we plan to expand on the technical challenges and proof introduced in the Appendix.
>
> > At the beginning of Section 4, there is a discussion on using two different KL-regularization coefficient during two stages. I am wishing to understand more about the choices and the reason behind. In particular, if I understand it correctly, using different beta essentially lead to different optimal values, which then damages the usefulness of the offline value estimation for the online phase.
>
> This is a great observation. Indeed, using different values of $\beta$ leads to different KL-regularized objectives and corresponding optimal value functions. Our design choice is driven by empirical findings.
>
> As $\beta \to \infty$, the KL-regularized objective becomes more conservative, and $V^\star(x)$ converges to the expected reward under the reference policy. Conversely, as $\beta \to 0$, $V^\star(x)$ approximates the $\text{Pass@}N$ performance of $\pi_{\text{ref}}$. The proofs are shown in Appendix E.
>
> We use a larger $\beta_1$ in the offline stage to estimate $V^\star(x)$ and a smaller $\beta_2$ in the online stage. Our intuition is that using a larger $\beta_1$ results in a more stable and realistic estimate of the achievable value after policy training, avoiding overly optimistic targets. During the online stage, we use a smaller $\beta_2$ to loosen the KL constraint, allowing the policy to better optimize reward. Appendix D presents an ablation showing that both overly optimistic or pessimistic estimates of $V^\star(x)$ can degrade final performance.
>
> > Regarding the theoretical innovation concern raised above, I am also looking forward to learning from the authors more about the underlying and probably hidden contributions there.
>
> We are grateful for your interest and are happy to elaborate on the theoretical contributions. Our analysis introduces several novel techniques and findings:
> - **Getting rid of policy completeness assumption.** In our analysis we only need policy realizability. In contrast, other on-policy policy optimization approaches including PG, PPO and REBEL rely on a stronger condition called policy completeness. We avoid such a condition by designing novel loss functions, which directly regress to the optimal policy rather than next-step policies similar to policy iteration.
> - **Getting rid of exploration mechanisms.** Our method does not incorporate any explicit exploration mechanism such as optimism in the face of uncertainty or reset to some exploratory distribution as in existing works [1, 2, 3, 4, 5] This is a significant improvement because exploration is necessary in general RL settings. We achieve this by noticing that in the context of LLMs, there exists a close connection between the novel loss functions we proposed and the KL divergence between the learned policy and the optimal policy (illustrated in step 2 of the proof in Appendix H). Leveraging this observation, we managed to bound the suboptimality of the learned policy efficiently.
> - **New findings on nonconvex no-regret learning with OGD.** In the case study of log-linear policy classes, we find that OGD provably converges with the optimal rate. This is a new finding because OGD doesn’t work for nonconvex losses in general. We achieve such a result by establishing a connection between the gradient of the loss and the distance from the optimal parameter (Lemma 3 in Appendix G). This connection enables us to efficiently bound the distance between the learned policy and optimal policy. Notably, the techniques used in Appendix G are totally different from the ones we use in Appendix H (Theorem 1), which is the reason why we can obtain a faster convergence rate.
>
> [1] Xie, T., Foster, D. J., Krishnamurthy, A., Rosset, C., Awadallah, A., & Rakhlin, A. (2024). Exploratory preference optimization: Harnessing implicit q*-approximation for sample-efficient rlhf. arXiv preprint arXiv:2405.21046.
>
> [2] Cen, S., Mei, J., Goshvadi, K., Dai, H., Yang, T., Yang, S., ... & Dai, B. (2024). Value-incentivized preference optimization: A unified approach to online and offline rlhf. arXiv preprint arXiv:2405.19320.
>
> [3] Zhang, S., Yu, D., Sharma, H., Zhong, H., Liu, Z., Yang, Z., ... & Wang, Z. (2024). Self-exploring language models: Active preference elicitation for online alignment. arXiv preprint arXiv:2405.19332.
>
> [4] Bai, C., Zhang, Y., Qiu, S., Zhang, Q., Xu, K., & Li, X. (2025). Online preference alignment for language models via count-based exploration. arXiv preprint arXiv:2501.12735.
>
> [5] Chen, M., Chen, Y., Sun, W., & Zhang, X. (2025). Avoiding $\mathbf {exp (R_ {max})} $ scaling in RLHF through Preference-based Exploration. arXiv preprint arXiv:2502.00666.

---

> > ### Comment · Reviewer_yXFw · 2025-08-05
> >
> > I would like to thank the authors for answering my questions, which have been largely resolved, and wish to maintain my favorable consideration of this work.

---

> > > ### Author Response · Authors · 2025-08-05
> > >
> > > Thank you for taking the time to read our rebuttal and maintain the rating. We will make sure to incorporate your suggestions into the revised version of the paper.

---

### Official Review · Reviewer_Wcpu · 2025-07-03

**Clarity:** 3
**Significance:** 2
**Originality:** 3
**Rating:** 4
**Confidence:** 4

**Summary:**

This paper proposes a way to significantly reduce the cost of online RL by decoupling value estimation. Specifically, for a given RL prompt dataset, they collect offline rollouts using a reference model to serve as the value baseline. Then during online RL, they use only 1 rollout per prompt and simplify the policy objective to a plain regression on the advantage (using MSE loss). Experiments are done on Qwen models and math domain.

**Questions:**

See the weaknesses part point 1.

**Ethical Concerns:**

["NO or VERY MINOR ethics concerns only"]

**Final Justification:**

The paper presents a simple yet effective method to decouple value estimation from online RL via offline rollouts with a reference policy. The idea is easy to implement and supported by theoretical grounding. While the scope of experiments is somewhat limited, the authors’ rebuttal provided useful clarifications and additional results on long-context models, which strengthen the case. I continue to view this as a technically solid contribution with efficiency benefits, though with limited breadth of evaluation. My borderline-accept rating remains unchanged.

**Limitations:**

yes

**Quality:**

2

**Strengths And Weaknesses:**

### Strengths:
- Easy to plug into existing pipelines. Compared to other methods that modify value or advantage estimators, regressing offline advantages is a fresh idea.
- The writing is clear and easy to follow.
- The theory section gives a solid in-principle analysis, which is nice to see.

### Weaknesses:
- The key novelty — using offline value estimation — needs more ablations. For example, I would really like to see how it performs if they just estimate value online with the same setup (i.e. just keep updating \pi in the process how will the value baseline change).
- Experiments are limited and focus only on short-context (non-thinking) models. I totally get the resource constraints (they do not even have complete baselines for thinking models), but it still undercuts the strength of the paper’s message.

---

> ### Author Rebuttal · Authors · 2025-07-29
>
> We sincerely thank the reviewer for the thoughtful and constructive feedback.
>
> > The key novelty — using offline value estimation — needs more ablations. For example, I would really like to see how it performs if they just estimate value online with the same setup (i.e. just keep updating \pi in the process how will the value baseline change).
>
> Our key contribution is to decouple value estimation from online learning by estimating the optimal value function offline using samples from the reference policy. This avoids the need for explicit critics or multi-sample estimation of the current policy’s value, which are required by PPO and GRPO and substantially increase both algorithmic complexity and training cost. While we understand the curiosity about online estimation under our setup, such an approach would revert to GRPO or PPO, thereby obscuring the benefits and novelty of our method. Moreover, as discussed in Section 2, the optimal value function under the KL-regularized RL objective is defined with respect to the reference policy $\pi_{\text{ref}}$, not the learned policy $\pi$. This provides a theoretical justification for our offline estimation approach and highlights the limitations of online estimation in this setting.
>
> > Experiments are limited and focus only on short-context (non-thinking) models. I totally get the resource constraints (they do not even have complete baselines for thinking models), but it still undercuts the strength of the paper’s message.
>
> We fully agree that evaluating on more complex, long-context reasoning tasks is important. To address this, we included experiments in Section 4.3 using the DeepSeek-R1 distilled Qwen-1.5B model with a 16k context length which is a setting for long-form reasoning. As shown in Table 2, $A^\star$-PO achieves the best performance on AIME 2025 and HMMT February 2024, and performs comparably to PPO and GRPO on AIME 2024 and HMMT February 2025. These results demonstrate that $A^\star$-PO effectively scales to long-context reasoning while remaining significantly more efficient in terms of training time and memory.

---

> > ### Comment · Reviewer_Wcpu · 2025-08-06
> >
> > I appreciate the authors’ clarifications. Also thanks for your explanations regarding novelty and experiments.
> > I will keep my positive score.

---

### Decision · Program_Chairs · 2025-09-17

**Decision:**

Accept (poster)

**Comment:**

This paper introduces A*-PO, a clever two-stage framework for fine-tuning LLMs on reasoning tasks using reinforcement learning. The core idea is to make the process more efficient by decoupling the costly value estimation from online training. It achieves this by first estimating the optimal value function offline using a reference model, and then updating the policy online with a simple regression loss that only requires a single generation per prompt. The method's main strengths are its impressive efficiency—significantly cutting training time and memory usage—and its elegant simplicity. The approach is novel, easy to implement, and well-supported by both solid theoretical guarantees and strong empirical results across various math reasoning benchmarks. While there were some initial questions from reviewers about the experimental scope and certain theoretical assumptions, the authors provided a thorough and convincing rebuttal that clarified these points well. This is a well-executed paper that presents a valuable and practical contribution to the field, and I'm happy to recommend its acceptance.